# The combined action of CTCF and its testis-specific paralog BORIS is essential for spermatogenesis

Samuel Rivero-Hinojosa [1,2✉], Elena M. Pugacheva [1✉], Sungyun Kang[1,3], Claudia Fabiola Méndez-Catalá[1,4], Alexander L. Kovalchuk [1], Alexander V. Strunnikov[5], Dmitri Loukinov[1], Jeannie T. Lee [6,7] & Victor V. Lobanenkov [1✉]

CTCF is a key organizer of the 3D genome. Its specialized paralog, BORIS, heterodimerizes with CTCF but is expressed only in male germ cells and in cancer states. Unexpectedly, BORIS-null mice have only minimal germ cell defects. To understand the CTCF-BORIS relationship, mouse models with varied CTCF and BORIS levels were generated. Whereas $Ctcf^{+/+}Boris^{+/+}$, $Ctcf^{+/-}Boris^{+/+}$, and $Ctcf^{+/+}Boris^{-/-}$ males are fertile, $Ctcf^{+/-}Boris^{-/-}$ (Compound Mutant; CM) males are sterile. Testes with combined depletion of both CTCF and BORIS show reduced size, defective meiotic recombination, increased apoptosis, and malformed spermatozoa. Although CM germ cells exhibit only 25% of CTCF WT expression, chromatin binding of CTCF is preferentially lost from CTCF-BORIS heterodimeric sites. Furthermore, CM testes lose the expression of a large number of spermatogenesis genes and gain the expression of developmentally inappropriate genes that are "toxic" to fertility. Thus, a combined action of CTCF and BORIS is required to both repress pre-meiotic genes and activate post-meiotic genes for a complete spermatogenesis program.

[1] Laboratory of Immunogenetics, National Institute of Allergy and Infectious Diseases, National Institutes of Health, Bethesda, MD, USA. [2] Center for Cancer and Immunology Research, Children's National Research Institute, Washington, DC, USA. [3] Department of Biology, Indiana University, Bloomington, IN, USA. [4] Laboratory of Genetics and Molecular Oncology, Building A4, Faculty of Higher Studies (FES) Iztacala, National Autonomous University of Mexico (UNAM), Tlalnepantla, State of Mexico, Mexico. [5] Guangzhou Institutes of Biomedicine and Health, Molecular Epigenetics Laboratory, Guangzhou, China. [6] Department of Molecular Biology, Massachusetts General Hospital, Boston, MA, USA. [7] Department of Genetics, Harvard Medical School, Boston, MA, USA. ✉email: sriverohin@childrensnational.org; epugacheva@niaid.nih.gov; vlobanenkov@niaid.nih.gov

How the three-dimensional (3D) genome is organized through diverse developmental stages is one of the key questions in the epigenetics field. The CCCTC-binding factor (CTCF) is a chief organizer of chromosome looping in multicellular eukaryotes ranging from insects to humans[1,2]. CTCF is an exceptionally conserved 11-zinc finger (ZFs) DNA-binding protein that binds thousands of genomic sites in the fly, mouse, and human genomes[3–5]. CTCF functions by halting cohesin-mediated extrusion of chromatin fibers thus creating chromosome loops[6–8]. CTCF thereby partitions the genome into autonomously functioning expression units[2,9]. Numerous nuclear processes including transcription regulation, imprinting, chromatin insulation, X-chromosome inactivation, and repair of DNA double-strand breaks (DSBs) have been linked to the organizational function of CTCF[10–21]. The essential function of CTCF is demonstrated by the early embryonic lethality of mice that are null for $Ctcf$[22]. Furthermore, although heterozygous $Ctcf$ knockout ($Ctcf^{+/-}$) mice are born at the expected Mendelian ratio and show no obvious developmental defects[22–24], the haploinsufficiency results in a predisposition to spontaneous tumorigenesis in a broad range of tissues[25]. Proper expression and function of CTCF are therefore essential both during development and in the adult soma.

The $Ctcf$ gene has a close paralog known as $Boris$ (Brother Of the Regulator of Imprinted Sites) or $Ctcfl$ (CCCTC-binding factor-like)[26,27] that appeared later in evolution as a duplication of $Ctcf$ in early vertebrates[28]. The two genes share a high degree of homology in the central 11-ZF DNA-binding domain. As a consequence, CTCF and BORIS proteins bind to virtually the same DNA motifs in vivo and in vitro[29–33]; however, they differ dramatically in their N- and C-terminal domains[34]. Recent studies show that CTCF and BORIS interact with different protein partners[35–37]. Notably, the divergence of the N-terminal region of the two proteins prevents BORIS from being able to block cohesin-mediated loop extrusion in contrast to CTCF[36,37].

Altogether, these observations suggest different developmental functions for the two paralogs. Indeed, whereas CTCF expression is ubiquitous, BORIS expression is normally restricted to male germ cells[27]. In mice, the testis is the only tissue affected to some extent by a $Boris$ knockout[32]. Nevertheless, $Boris$ knockout male mice have only minor spermatogenesis defects, including a reduction of testicular size due to increased apoptosis in germ cells and delayed production of haploid cells[29,31,32]. Surprisingly, $Boris$-null males are still fertile, though they exhibit subfertility.

Therefore, it is unclear at present what is the BORIS's exact molecular and developmental role and whether its close relationship to CTCF may account for the partial penetrance of $Boris$ KO in male germline. These questions are significant not only in a developmental context but also for the development of cancer. It is now well established that $Ctcf$ haploinsufficiency predisposes to cancer[25]. It is also known that BORIS is aberrantly activated in many types of cancer[30,35,38–42]. As such, BORIS has been recognized as a "cancer-testis antigen," but it is unclear whether it has a universal mechanism of action in cancer cells.

With respect to molecular interaction between the two factors, when CTCF and BORIS are co-expressed, either in germline or cancer cells, they bind at a subset of binding regions that contain at least two closely spaced CTCF-binding motifs (termed clustered CTCF target sites or 2xCTSes)[33,43]. In contrast to 1xCTSes bound only by CTCF, the 2xCTSes predispose the physical and heterodimeric interactions between CTCF and BORIS due to DNA-dependent constrains[33,44]. Interestingly, these heterodimeric sites of binding are enriched at active promoters and enhancers in both germ cells and cancer cells[33]. Moreover, the putative heterodimerization of CTCF and BORIS at the 2xCTSes have been shown to be involved in transcriptional program of

cancer cells and activation of testes-specific genes in cancer[33,43,45]. In mature spermatozoa, the 2xCTSes are associated with retained histones in mice and humans[33,46]. The exact degree of functional integration of CTCF–BORIS heterodimers currently remains unknown, as a systematic genetic analysis has yet to be performed in a normal developmental process—i.e., outside of cancer states.

To advance the understanding of both molecular and functional interaction between CTCF and BORIS, here we describe a genetic model in the male germline of the mouse—the only known normal context in which BORIS is expressed. Mouse spermatogenesis is a unique multistage process that occurs over a 2-week period, during which diploid spermatogonia are transformed into haploid germ cells. The dramatic transformation occurs morphologically, genetically, and transcriptionally, and therefore makes an excellent dynamic model in which to study CTCF–BORIS interactions. In this work, we generated mice with varying levels of CTCF and BORIS and investigated the functional consequences and effects on the spermatogenesis gene expression program. Our findings demonstrate that the CTCF–BORIS interaction is a developmental activator that is essential for induction of male gametogenesis genes.

## Results

**$Ctcf^{+/-}Boris^{-/-}$ male mice are infertile with spermatogenic defects and spermatozoa abnormalities.** Testis is the only mammalian tissue in which the two paralogous genes, $Ctcf$ and $Boris$, are normally co-expressed[27,41]. To examine the interplay between CTCF and BORIS during all stages of spermatogenesis, we generated mouse models in which the expression levels of CTCF and BORIS varied in testis. Because a complete $Ctcf$ deficiency results in embryonic lethality, we were only able to produce mice with heterozygous $Ctcf$ knockout (CTCF HET; $Ctcf^{+/-}$). In contrast, complete $Boris$ knockouts (BORIS KO; $Boris^{-/-}$) were viable. Thus, four mouse models were created for subsequent analysis: wild type (WT; $Ctcf^{+/+}Boris^{+/+}$), CTCF HET ($Ctcf^{+/-}Boris^{+/+}$), BORIS KO ($Ctcf^{+/+}Boris^{-/-}$), and compound mutant (CM; $Ctcf^{+/-}Boris^{-/-}$). Animals of each genotype were viable and indistinguishable from their WT littermates in survival rate and appearance. Breeding tests showed that mating behavior of all mouse models was within the norm, and the same frequency of copulation plugs was observed in all genotypes. However, only CM male mice displayed complete infertility (Table 1). Repeated breeding attempts to cross CM males with WT females yielded no offspring, with the exception of one stillborn deformed, five-legged, male pup (Supplementary Fig. 1a) that was twice the size of a normal newborn pup. A comparative genomic hybridization at the

**Table 1 Summary of in vivo phenotypes associated with different levels of CTCF and BORIS expression.**

| Mice | Genotype | Sex | Fertility |
|---|---|---|---|
| WT (wild type) | $Ctcf^{+/+}Boris^{+/+}$ | ♀ | YES |
| | | ♂ | YES |
| CTCF HET (CTCF heterozygote) | $Ctcf^{+/-}Boris^{+/+}$ | ♀ | YES |
| | | ♂ | YES |
| BORIS KO (BORIS knockout) | $Ctcf^{+/+}Boris^{-/-}$ | ♀ | YES |
| | | ♂ | YES (subfertility) |
| CM (compound mutant) | $Ctcf^{+/-}Boris^{-/-}$ | ♀ | YES |
| | | ♂ | NO |

For fertility analysis, 3-month-old animals were mated with either the same genotype or with WT. For example, WT and CM males were mated with WT or CM females and vice versa. No offspring were recovered with CM males. Normal sexual behavior for all genotypes was confirmed by the presence of vaginal plugs in all mated females.

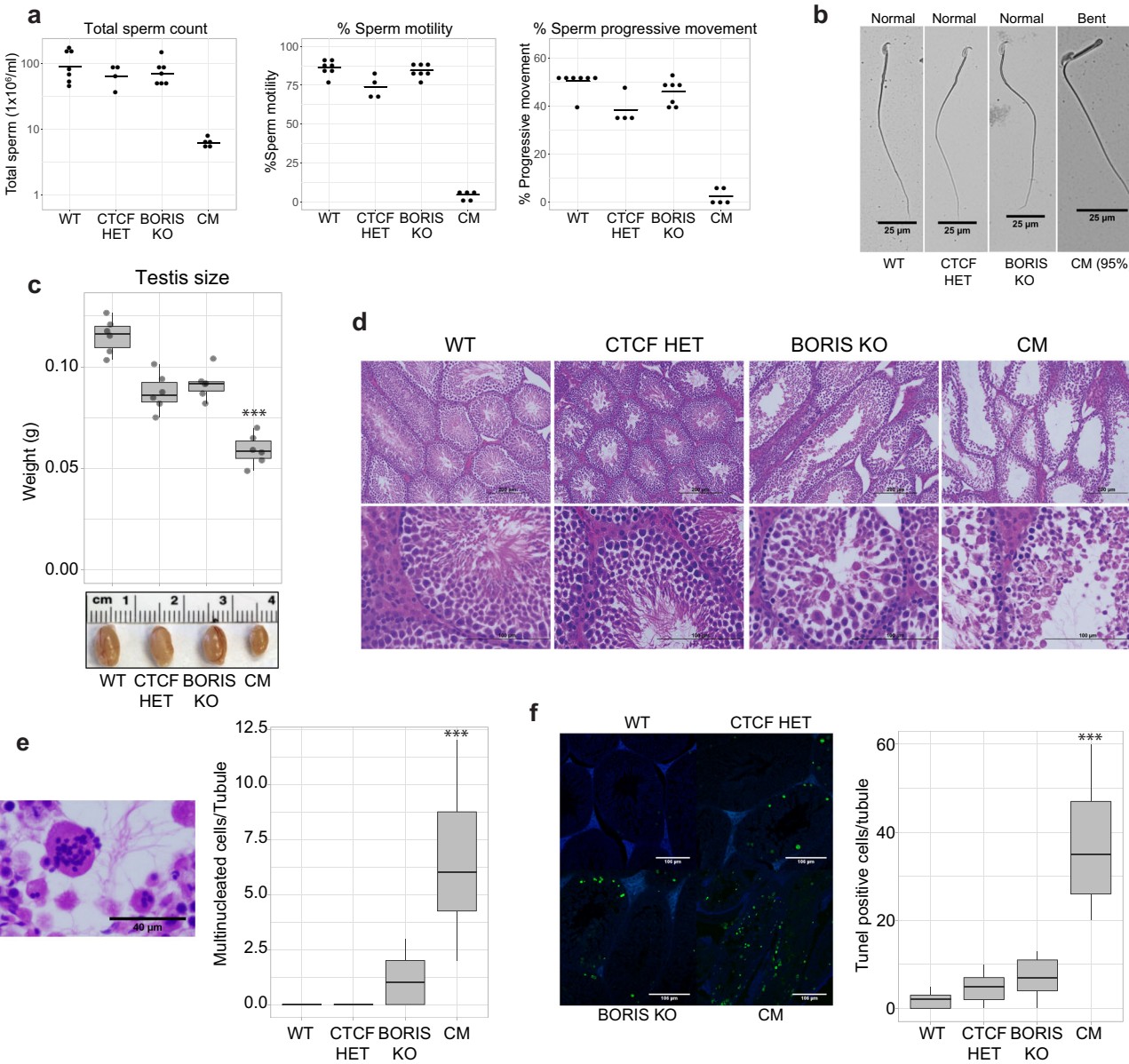

**Fig. 1 CM mice are infertile with spermatozoa abnormalities and spermatogenic defects. a** Sperm analysis in the four types of mice (WT; $n = 7$, CTCF HET; $n = 4$, BORIS KO; $n = 7$, and CM; $n = 5$ biologically independent animals). From left to right, counts of spermatozoa, percentage of motility, and progressive movement. The mean is indicated as a horizontal line. **b** Visualization by light microscopy of spermatozoa from WT, CTCF HET, BORIS KO, and CM mice. The image demonstrates the spermatozoa defects in sterile CM mice with tails bent 180° through the cytoplasmic droplet. Seven animals from each type of mice were analyzed with similar results. Scale bars: 25 μm. **c** Upper panel: box plot representing the testis weight (g) in the four types of mice (***p value = 2.825e−07, two-sided unpaired t test equal variance, compared to WT, $n = 6$). Lower panel: representative picture of testis obtained from four types of mice. **d** Hematoxylin and eosin staining of testis prepared from WT, CTCF HET, BORIS KO, and CM 3-month-old animals. Seven animals from each type of mice were analyzed with similar results. Scale bars: 200 μm (upper panels), 100 μm (lower panels). **e** Left panel: histological H&E staining of multinucleated cells present in CM seminiferous tubules. Scale bars: 40 μm. Right panel: box plot representing the quantification of multinucleated cells in the four types of mice (***p value = 1.14e−12, two-sided unpaired Mann–Whitney's test, compared to WT, $n = 30$, 10 tubules in 3 animals). **f** Left panel: representative examples of TUNEL staining in the testis from WT, CTCF HET, BORIS KO, and CM mice. Right panel: box plot representing the quantification of TUNEL-positive cells from the four types of mice (***p value = 3.39e−31, two-sided unpaired Mann–Whitney's test, compared to WT, $n = 90$, 30 tubules in 3 animals). Scale bars: 106 μm. In the box plots, the lower and upper hinges correspond to the first and third quartiles, the middle line indicates the median. The upper whisker extends from the hinge to the largest value no further than 1.5 times of the interquartile range (IQR). The lower whisker extends from the hinge to the smallest value at most 1.5 times of the IQR.

resolution of 6.5 kb revealed the presence of Y-chromosome material and no notable genomic abnormalities besides *Boris* heterozygosity in this pup.

Analysis of cauda epididymis showed greatly reduced sperm count in CM male mice (Fig. 1a and Supplementary Fig. 1b). In addition to reduced numbers, the CM sperm also manifested poor motility and progressive movement as compared to WT, CTCF HET, and BORIS KO mice (Fig. 1a). The vast majority of the CM spermatozoa were also morphologically abnormal, with approximately 90–95% of CM sperm having tails bent 180° through the cytoplasmic droplet and 3–5% having abnormal heads with a round shape (Fig. 1b and Supplementary Fig. 1c).

The examination of adult testes showed that testes from 3-month-old CM animals were significantly smaller than those of age-matched WT, CTCF HET, and BORIS KO mice ($p$ value = 2.825e−07; Fig. 1c). Histological analysis revealed that CM mice had reduced testicular cellularity compared to WT, CTCF HET, and BORIS KO (Fig. 1d). In contrast to WT, we observed a relatively large number of empty tubules in CM testes with reduced numbers of meiotic and post-meiotic cells, although some normal tubules were also present, indicating certain heterogeneity between tubules and animals (Fig. 1d and Supplementary Fig. 1d). Moreover, some of the CM tubules lacked elongated spermatids, suggesting a meiotic arrest at the spermatocyte stage. Furthermore, CM testes had numerous multinucleated giant cells (Fig. 1e and Supplementary Fig. 1d), a feature found in a number of spermatogenesis-defective mouse genotypes[47,48]. Similar multinucleated giant cells were also present in BORIS KO, CTCF HET, and WT testis but in much lower numbers (Fig. 1e). A test for apoptosis by terminal deoxynucleotidyl transferase dUTP nick end labeling (TUNEL) staining showed a dramatic increase in the average number of TUNEL-positive cells per seminiferous tubules in CM compared to WT, CTCF HET, and BORIS KO mice (Fig. 1f). Apoptosis appeared to occur at all steps of spermatogenesis in CM testes (Fig. 1f). The increased apoptosis in the testes of CM mice likely accounts for the reduced cellularity and size of the testis. Altogether, these data argue that the sterility of CM mice could be attributed to the combination of high apoptotic rates resulting in low sperm count and the reduced motility and abnormal morphology of sperm itself.

**$Ctcf^{+/−}Boris^{−/−}$ male mice show defects in meiotic recombination.** Meiosis is the key genetic process in the germline, as during meiosis I genetic diversity is generated by recombination between paired homologous chromosomes. Pairing of homologs is mediated by the synaptonemal complex (SC), a multiprotein meiosis-specific structure necessary for both synapsis and the completion of recombination during meiosis[49]. The increased apoptosis in spermatocytes, the absence of post-meiotic cells in some CM tubules, and the presence of multinucleated giant cells in CM seminiferous tubules (Fig. 1d–f) suggested an aberration at the early steps of spermatogenesis, such as during meiosis I. Surface spreads of spermatocytes isolated from 3-month-old WT and CM testes stained with antibodies specific for lateral (SYCP3) and central (SYCP1) SC components revealed obvious abnormalities in >40% of CM spermatocytes, not observed in spermatocytes from WT mice. Particularly, in WT pachytene spermatocytes, all autosomes were fully synapsed, shown by the contiguous and overlapping SYCP3 and SYCP1 signals along the entire length at pachynema (Fig. 2a and Supplementary Fig. 2a). In CM spermatocytes, although SYCP3 and SYCP1 signals were generally overlapping, SYCP3 also formed abnormal aggregates at some paired chromosomes (Fig. 2a and Supplementary Fig. 2a, shown by red arrows). These SYCP3 aggregates were observed either at the end or in the middle of paired chromosomes (Fig. 2a, b and Supplementary Fig. 2) and ranged from 0 to 6 per CM spermatocyte (Fig. 2a, b and Supplementary Fig. 2). At the same time, 60% of CM spermatocytes showed apparently normal SC formation (Supplementary Fig. 2b), indicating that the SYCP3 aggregation phenotype is either not fully penetrant or represents an accumulation of a transitional stage, not captured in WT mice.

The second independent molecular hallmark of meiosis I is programmed DNA DSBs introduced by endonuclease Spo11 at PRDM9-binding sites in leptonema[50]. DSBs are marked by a phosphorylated form of the variant H2AX histone (γ-H2AX)[51]. Indeed, both WT and CM spermatocytes showed accumulation of γ-H2AX signal in leptonema and zygonema as a part of normal progression of meiosis I (Fig. 2c). During late pachynema and diplonema, DSBs are repaired and consequently γ-H2AX remains only at unsynapsed sex chromosomes forming a meiotic sex vesicle[52]. Accordingly, we detected a specific accumulation of γ-H2AX staining only at the sex vesicle in WT pachytene spermatocytes (Fig. 2b, c). However, in CM pachytene spermatocytes, we observed an expansion of γ-H2AX signal beyond sex chromosomes, into autosomes (Fig. 2b, c, yellow arrows). The accumulation of γ-H2AX staining on autosomes of CM pachytene spermatocytes may indicate the persistence of unrepaired DSBs beyond early pachynema, as previously demonstrated for *Parp1* knockout[53] and for DNA repair factor Hus1 inactivation[54]. Nevertheless, the CM spermatocytes that reached diplonema showed normal γ-H2AX and SYCP3 staining similar to WT (Fig. 2c), suggesting that the pachynema spermatocytes with abnormal staining are preferentially eliminated from meiosis, possibly by apoptosis, or recover after a delay.

Taken together, our data indicate that the simultaneous depletion of both CTCF and BORIS in $Ctcf^{+/−}Boris^{−/−}$ male mice results in some meiotic defects, including abnormal formation of the SC and persistence of unrepaired DNA damage—further providing an underlying basis for the sterility phenotype of CM males.

**CTCF and BORIS co-bind at active promoters and enhancers in male germ cells.** We next assessed the underlying molecular mechanisms of simultaneous CTCF and BORIS depletion in male germ cells. It is logical to assume that the possibility of functional integration of two proteins is the highest in cell types, where their expression levels are comparable, as has been shown in cancer cell lines[33]. In testis, it is established that *CTCF* expression spans all stages of spermatogenesis, but reports of *BORIS* expression in mice and humans have been somewhat conflicting[31–33]. In situ hybridization of the human testis showed that *BORIS* is expressed in multiple isoforms in different stages of spermatogenesis from spermatogonia to round spermatids[30]. Single-cell transcriptomes of human testicular cells from puberty to adults[55] also demonstrate that *CTCF* and *BORIS* are co-expressed in most stages of spermatogenesis, with the highest levels of co-expression in spermatogonia and early spermatocytes (Supplementary Fig. 3a). To reconcile the diverse data, we reanalyzed single-cell RNA sequencing (RNA-seq) data of germ cells isolated from mouse testes[56] and also verified that *Ctcf* and *Boris* are co-expressed from early to the late stages of spermatogenesis in mice, as well as in humans. *Boris* expression was the highest in undifferentiated spermatogonia, decreased in spermatocytes, and upregulated again in post-meiotic spermatids (Fig. 3a).

Earlier studies had reported CTCF- and BORIS-binding sites in round spermatids and cancer cells[33]. There, CTCF and BORIS were found to co-bind and form heterodimers at a subset of genomic regions consisting of at least two closely spaced CTCF-binding sites (named 2xCTSes)[33]. Because we were interested in mapping binding sites across multiple developmental stages, including when BORIS levels are the highest, we performed chromatin immunoprecipitation–sequencing (ChIP-seq) in the whole testes of 3-month-old WT mice. Of note, spermatogonia have the greatest BORIS expression, but they are the rarest cells in testis and thus we could not perform ChIP-seq on this population of germ cells.

In an agreement with previous studies[33], ChIP-seq showed a selective binding pattern for CTCF (34,625 bound regions) and BORIS (7654 bound regions) (Fig. 3b, c). Only a minority of CTCF-bound CTCF target sites (CTSs) (6,257 of 34,625 or 18%)

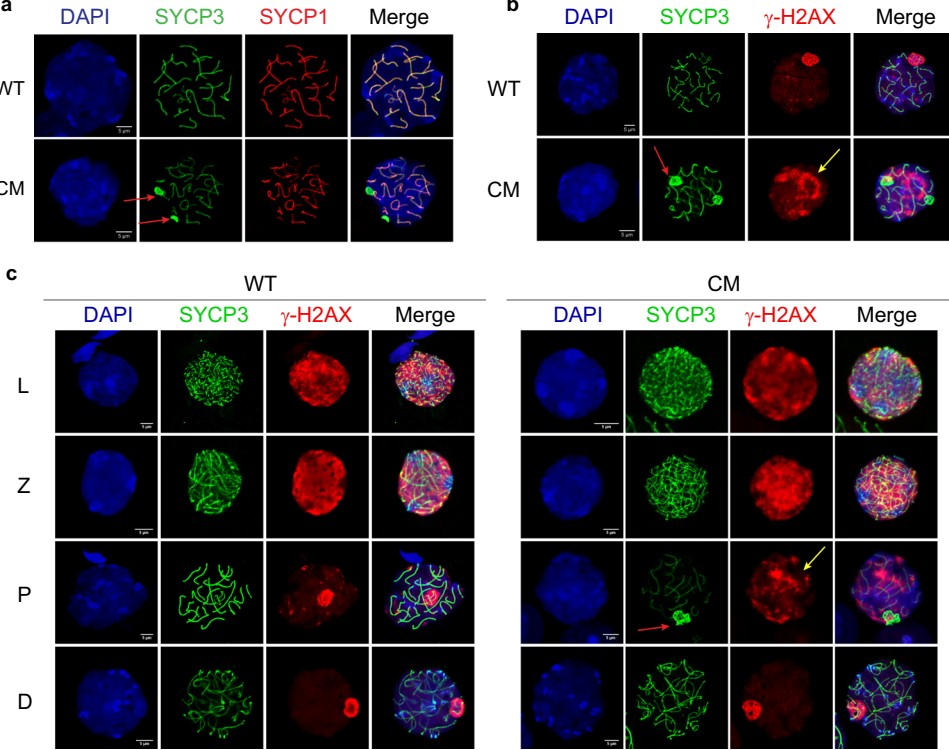

**Fig. 2 CM mice show defects in progression through meiosis.** Spermatocytes prepared from WT and CM mouse testes were fixed and analyzed by immunofluorescence microscopy. **a** Analysis of the synaptonemal complex in WT and CM testes. Spermatocyte spread preparations were stained with anti-SYCP3 (green) and SYCP1 (red) antibodies for visualization of the lateral and central synaptonemal complex components, respectively. The red arrows indicate abnormal SYCP3 staining. **b**, **c** Spermatocyte spread preparations were stained with anti-SYCP3 (green) and H2AX (red) antibodies. The red arrows show the abnormal SYCP3 staining of pachytene CM spermatocytes. The extension of H2AX staining beyond the sex vesicle in CM spermatocytes is shown by yellow arrows. Three animals from each type of mice were analyzed with similar results.

were co-occupied by BORIS (CTCF/BORIS sites), while the majority (28,368 of 34,625 or 82%) of CTSs were bound by CTCF alone (CTCF-only sites). Additionally, we mapped a small number of regions (1397) occupied by BORIS alone (BORIS-only sites). To analyze whether the regions bound by both CTCF and BORIS in mouse testis belong to the class of 2xCTS-binding sites, we carried out CTCF/BORIS motif analysis on the sequences around the summit of either CTCF/BORIS or CTCF-only and BORIS-only ChIP-seq peaks. Indeed, in contrast to 14% of CTCF-only binding sites, almost 70% of CTCF/BORIS-binding regions enclose at least two CTCF/BORIS-binding motifs that allow co-binding and heterodimerization of the two proteins with a similar 11-ZF DNA-binding domain (Supplementary Fig. 4a) as previously demonstrated[33]. Pathway analysis showed that a number of cellular processes are associated with the genes containing the three types of CTCF- and/or BORIS-binding sites (Supplementary Data 1).

In agreement with previous reports[33,57], CTCF/BORIS and BORIS-only binding regions were enriched in promoters, whereas CTCF-only regions were more enriched at intergenic and intragenic genomic regions (Fig. 3d). Furthermore, overlapping mouse testis ENCODE epigenomic data with our ChIP-seq data showed that BORIS binding (CTCF/BORIS and BORIS-only) was associated with active promoters enriched for the transcription initiating form of RNA polymerase II (RNAPII) and active histone modifications, H3K4me3 and H3K27ac, that mark active promoters and enhancers, respectively. These associations were at regions that are transcriptionally active during spermatogenesis (Supplementary Fig. 3b). To the contrary, CTCF-only regions were depleted of RNAPII and active histone marks in mouse testis (Fig. 3e). Thus, these data indicate an interaction of the two

paralogous proteins preferentially at active promoters and enhancers in the male germ cells.

**BORIS is required for full CTCF expression in male germ cells.** The complete sterility of CM, but not of CTCF HET or BORIS KO, male mice suggests that normal spermatogenesis proceeds with reduced CTCF levels in the presence of BORIS but is dysregulated when CTCF levels are reduced in the complete absence of BORIS. Incidentally, analysis of ChIP-seq data of CTCF and BORIS genomic occupancy in mouse testis showed that CTCF and BORIS bind their own promoters as well as each other's promoters (Fig. 4a), suggesting a possibility that CTCF and BORIS both autoregulate their own transcription and reciprocally regulate each other's transcription. We therefore analyzed *Ctcf* and *Boris* expression in testes of 3-month-old mice of the four genotypes by quantitative reverse transcriptase–PCR (qRT-PCR; Fig. 4b). As expected, *Ctcf* expression was reduced to 50% in CTCF HET testis compared to WT, where only one *Ctcf* allele is present (Fig. 4b). Surprisingly, *Ctcf* expression was significantly downregulated to approximately 70% of the WT expression level in BORIS KO testis in which both *Ctcf* alleles are present (Fig. 4b), suggesting that BORIS is necessary for the normal expression of *Ctcf*. Furthermore, the heterozygosity for *Ctcf* combined with *Boris* knockout in CM testis resulted in a further downregulation of *Ctcf* expression to only 25% of WT levels (Fig. 4b). Western blotting of CTCF protein levels in the testis and spleen from mice of the four genotypes corroborated the mRNA data (Fig. 4c). On the other hand, the deletion of the *Boris* gene had no effect on *Ctcf* expression in somatic tissues (Supplementary Fig. 5). Thus, the requirement for BORIS for full CTCF expression was observed only in testis, and the severity of

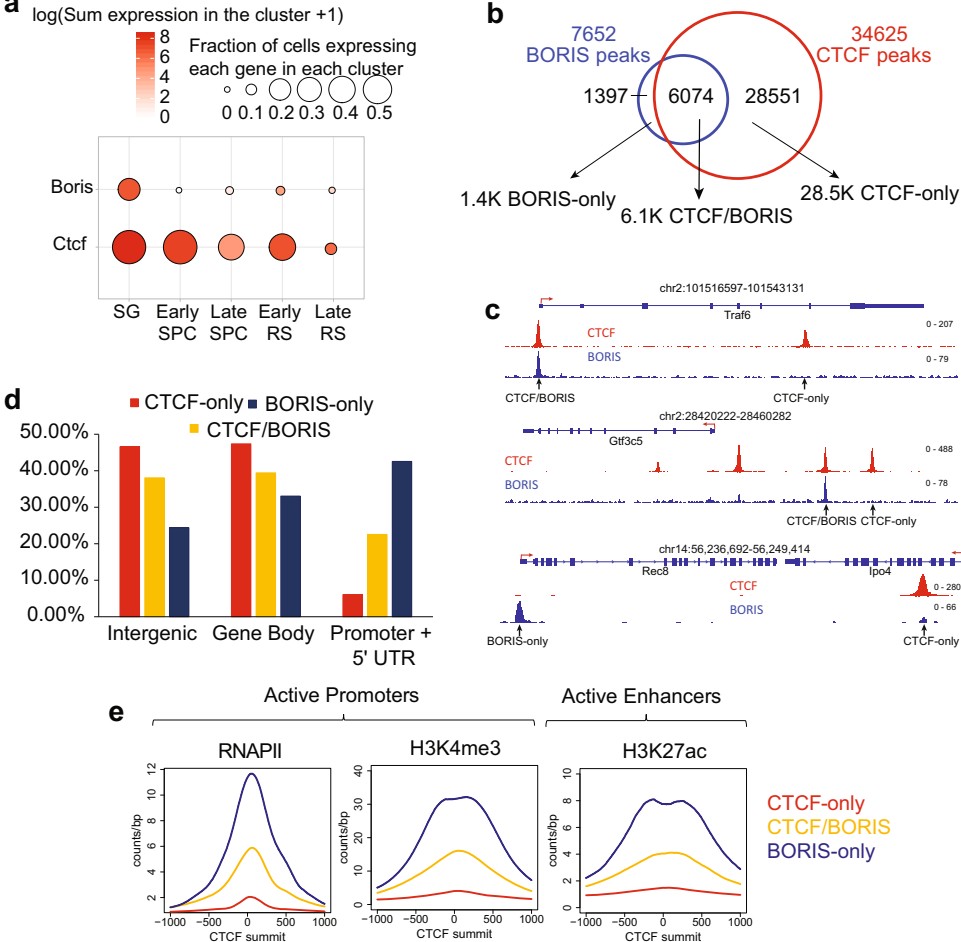

**Fig. 3 CTCF and BORIS cooperate at active promoters and enhancers in male germ cells. a** Single-cell RNA-seq analysis of *Ctcf* and *Boris* expression at the major stages of mouse spermatogenesis, SG spermatogonia, SPC spermatocytes, RS round spermatids. **b** Venn diagram depicting the overlap of CTCF (red) and BORIS (blue) occupancy in mouse male germ cells. Based on overlapping of CTCF and BORIS occupancy, 3 clusters were generated: BORIS-only, CTCF/BORIS, and CTCF-only. **c** Gene tracks of the distribution of CTCF (red) and BORIS (blue) bound regions in mouse male germ cells. **d** Barplot of the genomic distribution of CTCF and BORIS occupancy at promoters +5'UTR (3 kb upstream and downstream of TSS), gene bodies, and intergenic regions. **e** Average tag density of ChIP-seq data for RNA PolII (RNAPII), H3K4me3, and H3K27ac mapped in mouse testis (ENCODE data) across BORIS-only (blue), CTCF-only (red), and CTCF/BORIS (yellow) bound regions mapped in mouse testis.

CM phenotype may well be attributed to the combined effect of haploinsufficiency of *Ctcf* gene and the absence of BORIS-mediated upregulation of the remaining *Ctcf* allele.

**Preferential loss of CTCF from CTCF/BORIS dimeric sites in CM testes.** Given that CTCF expression decreased dramatically upon *Boris* knockout, we performed CTCF ChIP-seq to learn how CTCF binding to DNA changes in CM testes, where CTCF levels fell to 25% of normal. First, to determine whether the occupancy of CTCF-binding sites is maintained across different types of germ cells in mouse testis, we compared CTCF ChIP-seq in round spermatids and spermatocytes purified from 3-month-old mice with CTCF ChIP-seq from whole testis. We found that occupied CTCF-binding sites in whole testis are conserved in the individual types of germ cells: round spermatids and spermatocytes (Supplementary Fig. 6). This is not surprising, as CTCF occupancy is generally conserved across different cell types, with only ~20% of binding sites being cell type specific[58]. Thus, we proceeded with the mapping of CTCF-binding sites in germ cells isolated from the whole testes of WT and CM mice. In contrast to 34,625 CTCF ChIP-seq peaks identified in WT testis (Figs. 3b and 5a), only 21,525 CTCF peaks were identified in CM testis (Fig. 5a

and Supplementary Fig. 7a–c). At ~19,000 CTCF sites that were shared between CM and WT testes (Fig. 5a), CTCF peaks were 20% smaller in CM compared to WT (Fig. 5b). On the other hand, at 1922 sites the CTCF occupancy is actually increased in CM compared to WT testes (Fig. 5a and Supplementary Fig. 7a–c), which could be explained by either a slower CTCF turnover at these sites to compensate for the lower amount of available CTCF or by a necessity to maintain a high occupancy at these CTCF sites. Interestingly, we have not observed any significant loss of CTCF occupancy in BORIS KO germ cells, suggesting that the level of CTCF in BORIS KO is functionally sufficient to keep CTCF-binding pattern close to that in WT mice (Supplementary Fig. 6d).

As the male sterility phenotype emerged only in CM mice, we also asked whether the loss of CTCF occupancy was specifically associated with CTCF/BORIS sites. *K*-means ranked clustering of CTCF and BORIS occupancy along 15,198 lost CTCF-binding sites showed a large cluster of CTCF/BORIS sites that lost CTCF occupancy in CM compared to WT testes (Supplementary Fig. 7d). This, in turn, prompted us to analyze CTCF occupancy at CTCF-only and CTCF/BORIS sites in WT and CM testes (Fig. 3b). The average plot analysis of normalized tag density

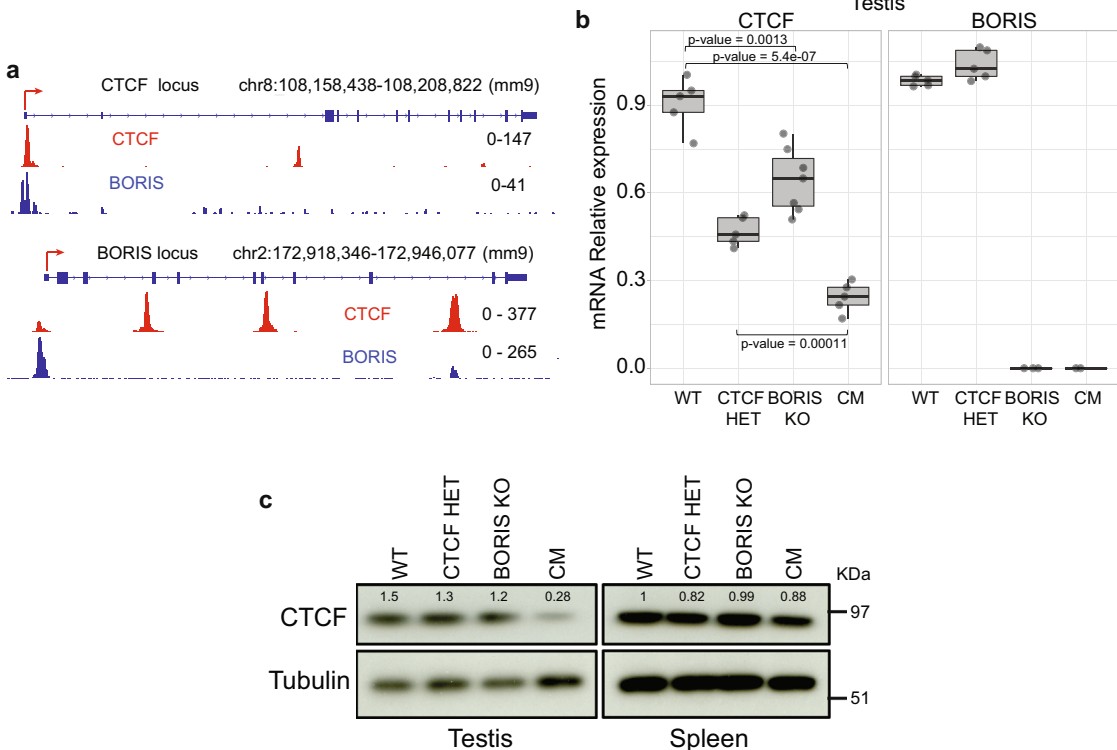

**Fig. 4 BORIS positively regulates *Ctcf* gene expression in testis. a** Genome browser view of CTCF (red) and BORIS (blue) occupancy at the promoter regions of the *Ctcf* and *Boris* genes in WT testes. **b** Expression of *Ctcf* and *Boris* analyzed by real-time PCR in testis of the four mouse genotypes. All data were normalized to *Gapdh*. Two-sided unpaired *t* test was performed; the *p* values are indicated in the figure (WT, CTCF HET, and CM; $n = 5$, BORIS KO; $n = 7$). The lower and upper hinges correspond to the first and third quartiles, the middle line indicates the median. The upper whisker extends from the hinge to the largest value no further than 1.5 times of the interquartile range (IQR). The lower whisker extends from the hinge to the smallest value at most 1.5 times of the IQR. **c** Western blot analysis of CTCF protein levels in the testis and spleen from the four types of mice. Tubulin was used as a loading control. Two independent experiments were repeated with similar results. Densitometry values were normalized to tubulin.

showed 25% reduction of CTCF occupancy at CTCF-only sites (*p* value: 7.321477e−27, Wilcoxon test) and a much more severe loss (50%) at CTCF/BORIS sites (*p* value: 2.175676e−66, Wilcoxon test) (Fig. 5c). Additionally, some CTCF/BORIS sites lost CTCF occupancy completely in CM (Fig. 5d). Thus, CTCF binding is preferentially lost from CTCF–BORIS dimeric sites, which are associated with active transcription in the male germ cells.

To further assess the specificity of CTCF-binding loss in CM testis, we combined CTCF peaks mapped in both WT and CM testes into a composite set of CTCF-binding sites (36,547) and separated them into three groups based on the differential CTCF ChIP-seq tag density as follows: WT > CM sites with the tag density more than twofold higher in WT than in CM (13,385 CTCF peaks), WT < CM sites with greater than twofold lower CTCF occupancy in WT than in CM (2,819 CTCF peaks), and CTCF = CM sites where tag density did not change more than twofold (7,051 CTCF peaks) (Supplementary Fig. 7e, f). An analysis of BORIS occupancy in WT testes at CTCF-binding regions with differential WT/CM CTCF occupancy demonstrates that the loss of CTCF binding in CM is preferentially associated with CTCF/BORIS-binding regions, as WT > CM sites were at least twofold enriched with BORIS occupancy compared to WT = CM sites (Supplementary Fig. 7g). These results also suggest that binding at CTCF/BORIS heterodimeric sites has greater stability than CTCF binding alone. An analysis of genomic distribution of differentially bound CTCF regions indicated that the CTCF binding in CM was preferentially lost (WT > CM) at the promoter regions compared to CTCF-binding sites that remained bound in CM cells (WT < CM and WT = CM;

Supplementary Fig. 7h). Moreover, CTCF-binding sites that lost CTCF occupancy in CM (WT > CM) were highly associated with transcription regulatory regions as compared to WT < CM or WT = CM regions, as indicated by the higher enrichment of ChIP-seq tag density of the active transcriptional marks, RNAPII, H3K4me3, and H3K27Ac, specifically at WT > CM regions (Fig. 5e). Thus, the lost CTCF-binding sites in CM testis are predominately associated with BORIS binding and active transcription marks in WT testes.

**Loss of spermatogenesis-specific gene expression in CM germ cells.** Considering the severe spermatogenesis defects in CM germ cells (Figs. 1 and 2), we used RNA-seq to analyze underlying gene expression changes in the testes of our four mouse models at 3 months of age. Significant changes were indeed observed in CM testis. CTCF HET and BORIS KO showed only 0 and 58 differentially expressed genes (DEGs), respectively (Fig. 6a, b, fold change >2, adjusted *p* value < 0.001, Supplementary Data 2), consistent with minimal effects on fertility in these animals. By contrast, CM testis showed 1,513 DEGs relative to WT (Fig. 6a, b and Supplementary Data 2). RNA-seq results were validated by qRT-PCR on a subset of DEGs (Supplementary Fig. 8). DEGs could either be upregulated or downregulated, as it is usually observed in such transcriptome-wide analyses, consistent with these being both direct and indirect targets. However, >400 genes were downregulated by the loss of *Boris* and haploinsufficiency of *Ctcf*, encompassing of 26% of all DEGs in CM testis.

It is evident, however, that differential gene expression detected in a tissue as a whole could be affected by changing cell-type

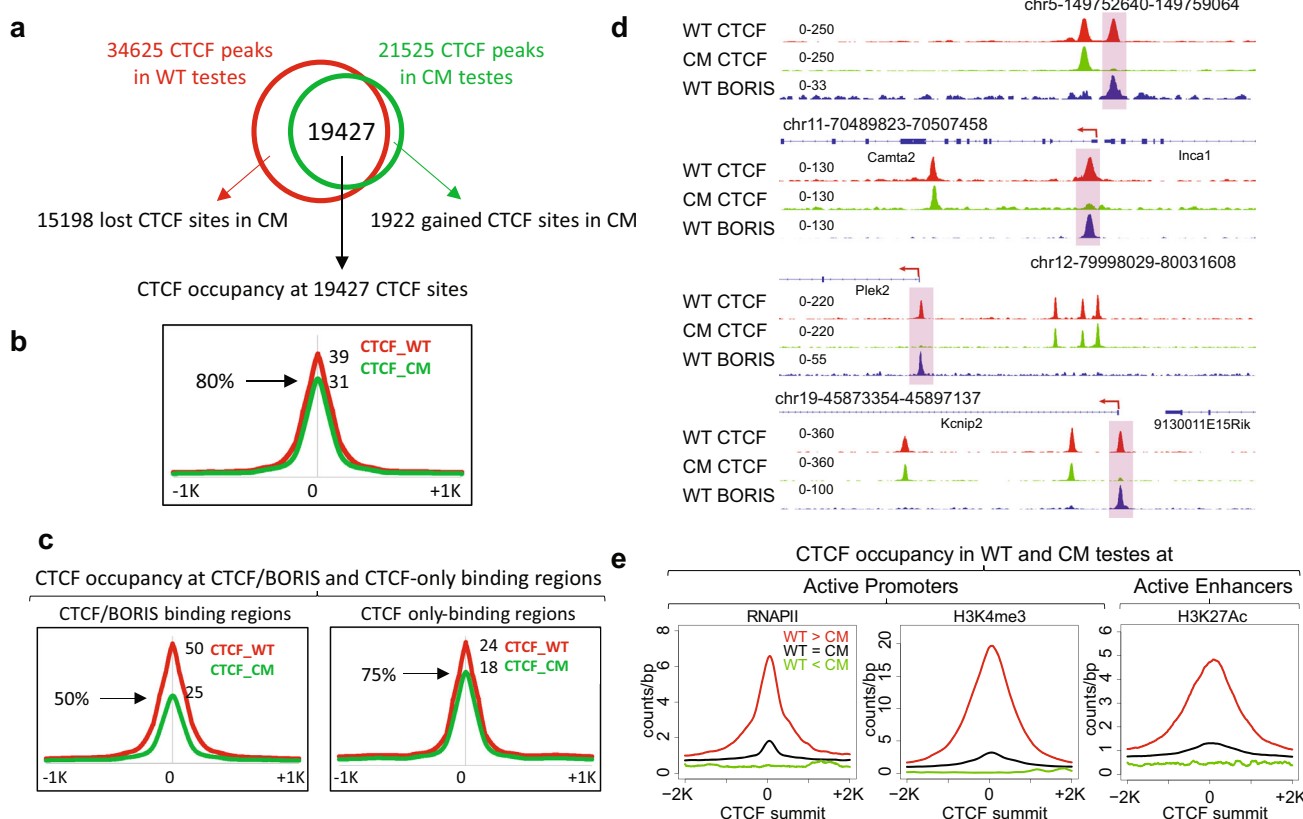

**Fig. 5 The loss of CTCF binding in CM testes occurs preferentially at CTCF/BORIS heterodimeric sites. a** Venn diagram shows an overlapping of CTCF ChIP-seq peaks mapped in WT (red) and CM (green) testes. **b** Average tag density of CTCF occupancy in WT and CM testes at 19,427 CTCF-binding sites. The black arrow shows connection between **a** and **b**. **c** Average tag density of CTCF occupancy mapped by ChIP-seq in WT (red) or CM (green) male germ cells across CTCF/BORIS (left panel) and CTCF-only (right panel) bound regions mapped in WT mouse testis. **d** Gene tracks for specific locations on mouse chromosomes 5, 12, 11, and 19 showing the distribution of CTCF (red) and BORIS (blue) bound regions in WT and CTCF-bound regions in CM (green) mouse male germ cells. Highlighted rectangles indicate the loss of CTCF occupancy in CM male germs cells (preferentially at CTCF/BORIS-bound regions mapped in WT testis). **e** Differentially bound CTCF regions are associated with active chromatin marks in WT germ cells. Average tag density of RNA PolII (RNAPII), H3K4me3, and H3K27ac mapped by ChIP-seq in mouse testis (ENCODE data) across regions differentially bound by CTCF (WT > CM, WT = CM, and WT < CM, loss/decrease of CTCF binding in CM compared to WT, no differences in binding, and preferentially bound in CM, respectively).

composition. Incidentally, whole testis contains multiple populations of germ cells that diverge in their stages of growth and differentiation. To ensure that the observed changes in gene expression program were not due to the potential changes of cell-type composition of the CM tubules, we performed RNA-seq analysis of round spermatids and spermatocytes purified from the four mouse genotypes. To validate the enrichment of specific germ cell types used for the RNA-seq analysis, we compared the transcriptional profiles of round spermatids and spermatocytes with a set of previously described markers[59] and also selected 833 post-meiotic genes from our analysis, which are highly expressed in round spermatids, but not in spermatocytes. Supplementary Fig. 9 demonstrates that the expression of round spermatids markers not only distinguishes meiotic (spermatocytes) and post-meiotic (round spermatids) cells in the four mouse genotypes but also shows that those markers are not significantly affected in CM round spermatids compared to other types of mice. Analyzing RNA-seq data from purified round spermatids and spermatocytes of the four mouse genotypes, we confirmed the non-additive impact of *Ctcf* haploinsufficiency joined with BORIS depletion on the deregulation of multiple genes in CM mice (Fig. 6b, c). Most importantly, we show that the DEGs in CM whole testis are deregulated in the same direction in CM round spermatids and spermatocytes, but to a lesser extent in the latter (Fig. 6b).

Reciprocally, DEGs in CM round spermatids are also deregulated in the same direction in whole testis (Fig. 6c), confirming that the RNA-seq analyses of both the individual germ cell types and whole testis are valid and complementary to each other. Thus, we suggest that the massive transcriptional deregulation observed in whole testis and in the main germ cell types of CM mice is a consequence of the simultaneous depletion of both CTCF and BORIS proteins.

Gene ontology (GO) analysis revealed that the downregulated genes in CM whole testis were significantly associated with sexual reproduction, spermatogenesis, male gamete generation, spermatid differentiation, and fertilization (Fig. 6d and Supplementary Data 3). By contrast, upregulated genes were not associated with a uniform pathway (e.g., oxidation/reduction, blood vessel development, lipid biosynthesis process, immune response, apoptosis; Fig. 6d and Supplementary Data 4). To confirm the effect of combined CTCF and BORIS depletion at pathway level, we performed a Gene Set Enrichment Analysis (GSEA) comparing CTCF HET, BORIS KO, and CM testes to WT testis. The top 10 downregulated pathways related to spermatogenesis in CM testis were also analyzed in CTCF HET and BORIS KO testes. As indicated in Supplementary Fig. 10, the normalized enrichment scores (NES) were significantly higher in CM compared to CTCF HET and BORIS KO, confirming the non-additive impact on the

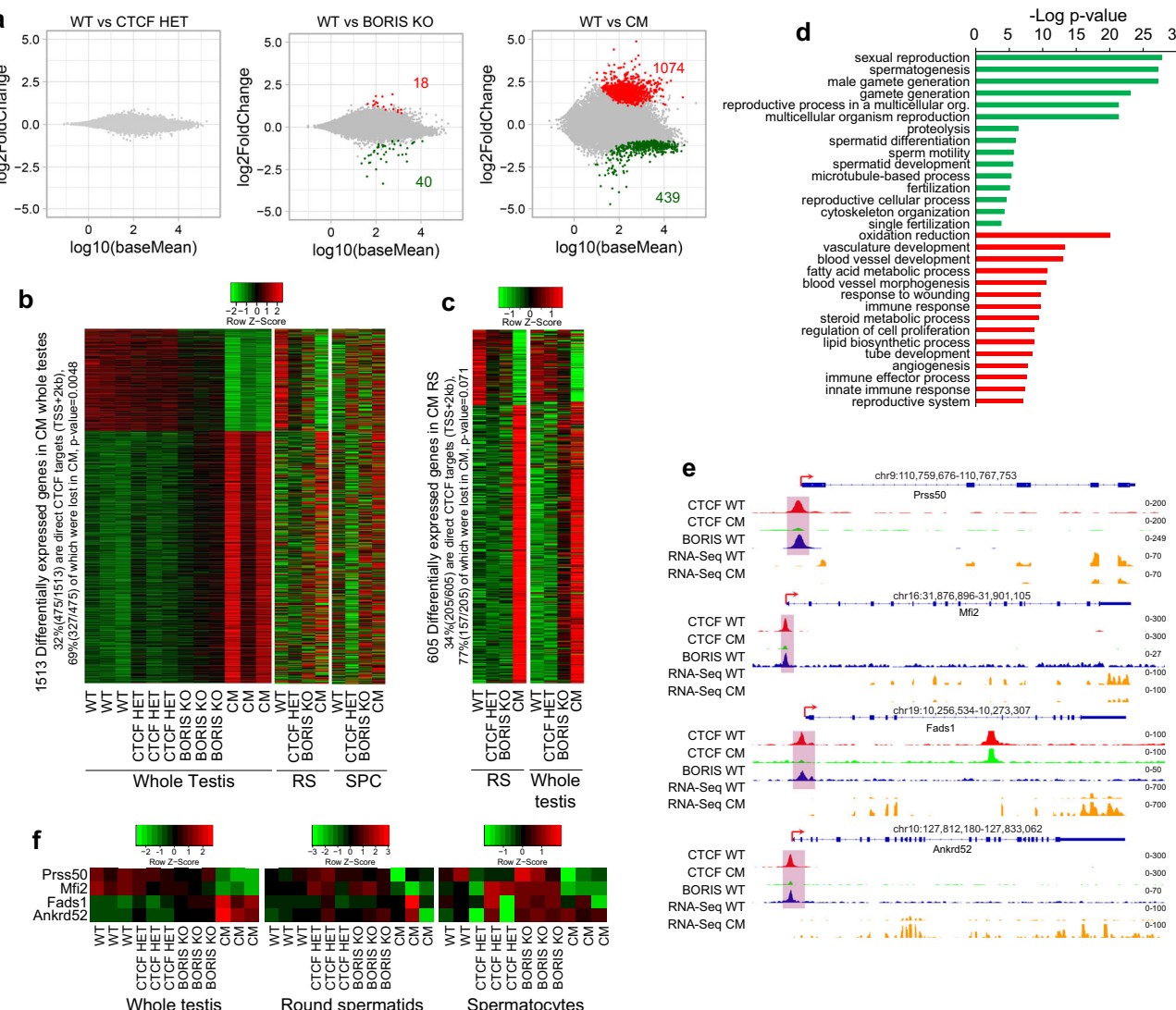

**Fig. 6 Synergistic effect of CTCF and BORIS deletion on spermatogenesis-specific gene expression. a** MA plots of RNA-seq in whole testis for CTCF HET, BORIS KO, and CM mice. The *y*-axis represents the log2 of fold change compared to WT; the *x*-axis represents log10 of mean expression. The red and green dots indicate the upregulated and downregulated genes, respectively. The gray dots indicate genes without significant changes. The number of differentially expressed genes (DEGs) are highlighted in red (fold change >2 and FDR < 0.001). **b** Heat map of DEGs in CM whole testis compared to round spermatids (RS) and spermatocytes (SPC) in the four types of mice; three replicates for each type of mice are represented for whole testis and average of the replicates for RS and SPC. **c** Heat map of DEGs in CM RS compared to whole testis; average of the replicates is shown. **d** Gene ontology analysis of downregulated (upper panel, green color) and upregulated (lower panel, red color) genes in CM testis compared to WT. The analysis was performed in DAVID Bioinformatics Resources 6.8 (https://david.ncifcrf.gov); Fisher's exact *p* values from the DAVID software are shown. **e** Genomic tracks displaying CTCF (in WT (red) and CM (green) testes) and BORIS (WT in blue) occupancy in comparison with RNA-seq data (yellow) in WT and CM germ cells across *Prss50* (*Tsp50*), *Mfi2*, *Fads2*, and *Ankrd52*. Red arrows indicate the TSS. **f** Heat map showing the change of expression of the same genes in whole testis, RS, and SPC.

pathways associated with spermatogenesis. Thus, there was a general negative effect on the spermatogenesis gene expression program in CM testis, suggesting that BORIS and CTCF must normally cooperate to regulate the spermatogenesis program.

Among the downregulated DEGs in CM are some confirmed direct targets of CTCF and BORIS, such as *Cst* (*Gal3st1*), *Tsp50* (*Prss50*), and *FerT* (*Fert2*)[29,32,60]. These genes were not only downregulated in whole testis but also in such individual germ cells types as round spermatids and spermatocytes (Supplementary Fig. 11a). The combined effect of simultaneous CTCF and BORIS depletion was even more obvious comparing DEGs in CM to BORIS KO testes, with the majority of DEGs being affected in a similar way but more profoundly in CM compared to BORIS KO testes (Fig. 6b). Notably, the most highly downregulated and

upregulated genes in CM testis compared to WT were also downregulated and upregulated in BORIS KO testis (Fig. 6b and Supplementary Data 2). These data demonstrate that CTCF expression can partially compensate for the loss of BORIS in the BORIS KO, but when CTCF expression is reduced in CM mice, it results in a much stronger impact on the spermatogenesis transcription program. These dramatic changes could therefore explain the infertility of CM males.

To ask whether altered CTCF/BORIS occupancy in CM testis could be the basis of differential gene expression, we integrated ChIP-seq and RNA-seq data for WT versus CM. Significantly, 37% (557 genes out of 1,513, Fisher exact test: *p* value = 0.0001) of DEGs in CM testis had CTCF or/and BORIS binding within or in close proximity to the genes (±2 kb) in WT testis

(Supplementary Fig. 11b and Supplementary Data 5). This association suggested that the loss of CTCF/BORIS binding might underlie the differential expression. Therefore, we examined some previously described CTCF/BORIS target genes. For example, *Prss50* harbors a CTCF and BORIS heterodimeric site (2xCTS) in the promoter region[29]. In BORIS KO testis, we observed a ~1.5-fold downregulation of *Prss50* expression; by contrast, we found >5-fold downregulation of *Prss50* expression in CM testis versus WT testis (Fig. 6e, f and Supplementary Fig. 11a). Similar patterns were observed for other genes including *Mfi2*, with a severe downregulation of expression in CM testis (Fig. 6e, f and Supplementary Fig. 12a, b).

Overall, among all DEGs (1,513), 475 (32%) genes were bound by CTCF in the promoter region (transcription start site (TSS) +2 kb) in WT cells, and almost half of these (69%, 327 genes, Fisher exact test, *p* value = 0.0048) showed a loss of CTCF binding in CM cells (Supplementary Data 5), pointing to a direct impact of CTCF binding on gene transcription in testis. Similarly, among 605 DEGs found in the round spermatids of CM testis, 205 (34%) genes were bound by CTCF in the promoter region (TSS +2 kb) in WT cells, and 157 of them (77%, Fisher exact test, *p* value = 0.071) showed a loss of CTCF binding in CM cells (Fig. 6c). To further refine the GO analysis of DEGs, we sorted the genes based on CTCF and BORIS occupancy and with respect to the loss of CTCF occupancy in CM testis. As shown in Supplementary Fig. 13a, the spermatogenesis-related pathways were significantly downregulated upon loss of CTCF occupancy either at CTCF/BORIS or CTCF-only sites, while all upregulated pathways were not related to spermatogenesis and were equally enriched independent of CTCF and BORIS occupancy.

DEGs in CM testis were strongly correlated with the enrichment of RNAPII and H3K4me3 histone modification as shown by ChIP-seq in germ cells isolated from WT and CM testes (Supplementary Fig. 13b). To integrate the transcriptomic and epigenomic datasets, DEGs were separated into three groups: downregulated, upregulated, and unchanged transcription for WT versus CM, and tag density plots of RNAPII and H3K4me3 ChIP-seq were generated for each group. The downregulation or upregulation of DEGs was well correlated with tag density for RNAPII and H3K4me3 around the TSS (Supplementary Fig. 13b). Although these results present a coherent interpretation of both RNA-seq signal and RNAPII or H3K4me3 occupancy at the promoters of DEGs affected by lost CTCF and BORIS binding in CM testis, the loss of CTCF binding is unlikely to explain all observed transcriptional changes in CM testis. We cannot exclude the possibility that the deregulation of some genes in CM testis with no loss of CTCF occupancy in the promoter regions is related to small changes in germ-cell populations due to the heterogeneity found in CM testis (Supplementary Fig. 1d).

Taken together, our findings demonstrate that both CTCF and BORIS activate the transcriptional program of spermatogenesis in male germline. Therefore, the loss of CTCF and BORIS occupancy specifically at CTCF/BORIS sites at the promoters of DEGs is likely the main contributing factor to the sterility of the CM male mice.

**"Toxic" KIT persistence beyond spermatogonia in the absence of CTCF and BORIS binding**. As mentioned above, transcriptomic analysis showed that a minority of genes was downregulated in CM testes. For example, *Fads1* and *Ankrd52* were highly upregulated in CM testes compared to WT upon loss of CTCF and BORIS occupancy in their promoter regions (Fig. 6e, f). Although GO analysis suggests indirect effects on some of these genes (Fig. 6d), the resulting overexpression of others could contribute to the sterility phenotype. In general, for other highly

DEGs, the loss of specific CTCF/BORIS-binding sites resulted in a severe deregulation of their transcription (Supplementary Fig. 12b, c).

Among the genes affected in CM testis, one gene of particular biological importance was upregulated: the proto-oncogene *Kit*, a member of the tyrosine kinase receptor family that is crucial for the survival, proliferation, and maturation of primordial germ cells[61]. Although *Kit*-deficient mice die in the first days of postnatal life, some mutations in *Kit* result in viable but sterile male mice[61]. In testis, *Kit* is expressed in two isoforms: before meiosis, it is expressed as a full-length transcript (*Kit FL*); after meiosis, it is expressed as a truncated isoform (*Kit TR*)[61]. KIT is routinely used as a marker of differentiating spermatogonia, as *Kit* expression is strictly restricted to these cells, and its expression is necessary for triggering the differentiation process in the noncommitted spermatogonia stem cells[61]. Persistence of high-level KIT could potentially disrupt downstream differentiation of the developing germ cells, as an earlier study showed that constant expression of a *Kit* mutant in spermatids caused malformed mature spermatozoa[62]. Incidentally, the *Kit* promoter contains a CTCF/BORIS heterodimeric site, and in CM testis, the loss of both CTCF and BORIS occupancy resulted in a significant *Kit* upregulation (Fig. 7a and Supplementary Fig. 14a). Furthermore, qRT-PCR validation showed that *Kit FL* transcript expression was increased at least 12-fold in CM testis (Fig. 7b), but the expression of *Kit TR* isoform was unaffected (Fig. 7c). An increase in the overall level of Kit in CM testis was also detected by immunoblotting (Fig. 7d). Accordingly, GSEA showed significant upregulation of the 52-gene KIT pathway, with a NES of 1.52 and a nominal *p* value of 0.024 (Supplementary Fig. 14b). Thus, *Kit* belongs to a group of DEGs that become upregulated in CM testis upon depletion of both CTCF and BORIS proteins in CM testis.

To determine whether the induction of *Kit* expression was a direct result of CTCF and BORIS simultaneous depletion, we first narrowed down the promoter-proximal region bound by CTCF and BORIS in WT testes to the first intron of mouse *Kit*, approximately 800 bp downstream of the TSS (Fig. 7a). This region was previously identified as an essential regulatory element for *Kit* expression[63]. Second, by gel-shift assays, we confirmed that both CTCF and BORIS bound equally well to the *Kit* regulatory region (Fig. 7e). The sequence of this region encompassed two 20-bp CTCF consensus sites separated by a 31-bp sequence (Fig. 7f), thus belonging to the 2xCTS class described previously[33]. This result is in good agreement with the simultaneous occupancy of CTCF and BORIS proteins at the same binding region (Fig. 7a). The presence of a 2xCTS under one ChIP-seq peak also explains the double shift of DNA–protein complex observed by the gel-shift experiments using the 11-ZF CTCF domain for the binding analysis (Fig. 7e, double arrow). Thus, CTCF and BORIS co-binding to the promoter region of *Kit* directly represses *Kit* transcription during spermatogenesis.

Comparison of gene regulation among evolutionarily distant genomes permits identification of evolutionary conserved mechanisms in gene regulation. To investigate whether the mechanism of *KIT* regulation is conserved in humans, we analyzed *KIT* expression in the BORIS-positive human erythroleukemia cancer cell line, K562. ChIP-seq mapping of CTCF and BORIS occupancy in K562 cells[33] showed that the same region of the *Kit* locus is occupied by both CTCF and BORIS (Fig. 7g). The alignment of mouse and human sequences under the CTCF/BORIS ChIP-seq peaks showed that both CTCF-binding sites are conserved, confirming an evolutionary importance of CTCF/BORIS binding for *KIT* transcription (Fig. 7f). To analyze whether CTCF and BORIS binding can also regulate *KIT* transcription in human cells, we knocked down BORIS

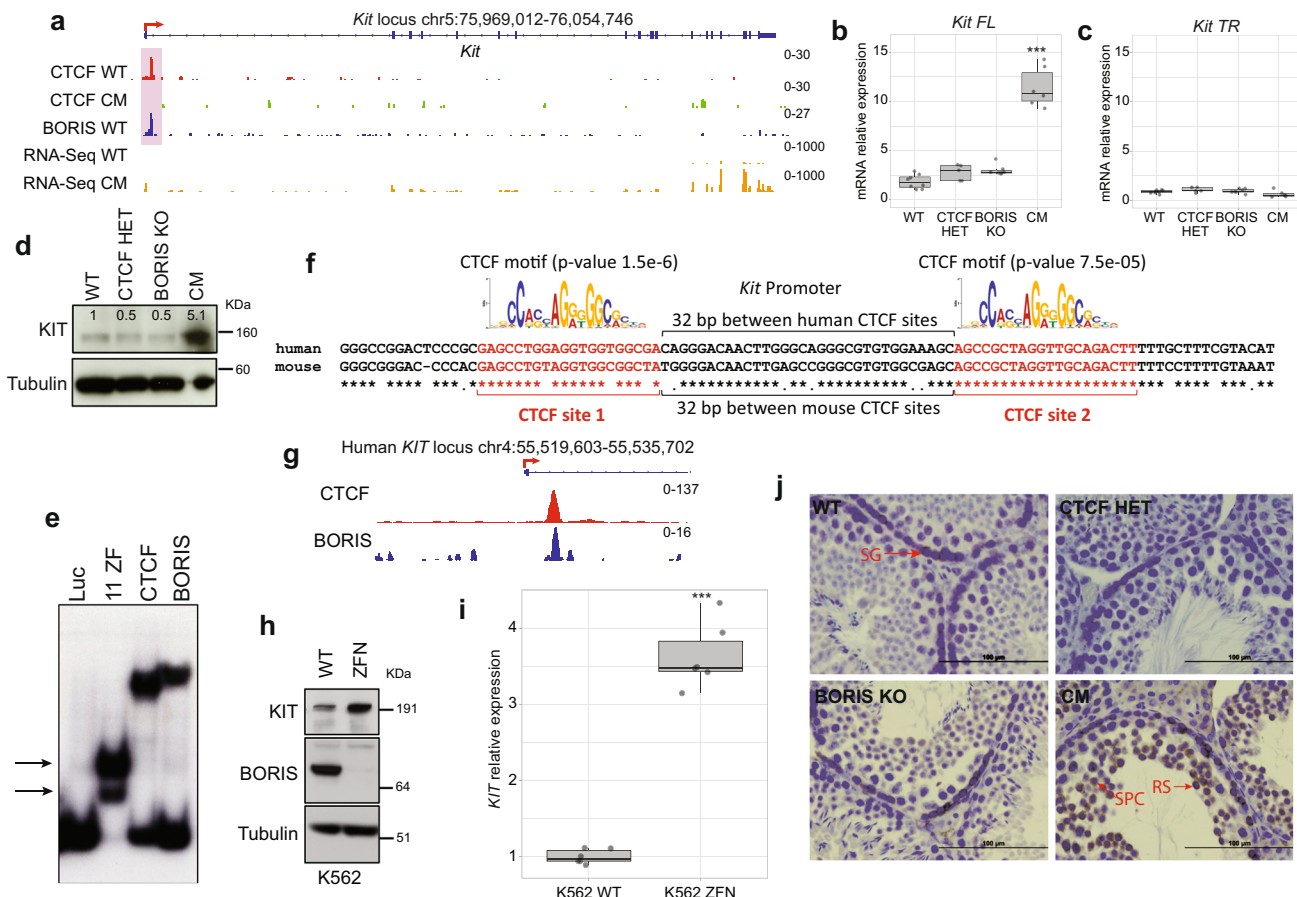

**Fig. 7 Depletion of BORIS and CTCF expression is associated with toxic Kit persistence. a** Genomic views of CTCF ChIP-seq in WT (red) and CM (green) testes, BORIS ChIP-seq in WT (blue), and RNA-seq (yellow) in WT and CM testes across the *Kit* locus. **b, c** Expression of full-length (FL) and truncated (TR) *Kit* transcript by qPCR in the four types of mice (***p value = 2.93e−08 for *Kit FL*, p value = 0.1 for *Kit TR*, two-sided unpaired *t* test, compared to WT, WT; n = 8, CTCF HET; n = 4, BORIS KO; n = 7, CM; n = 6). **d** Western blot of KIT in the four types of testis lysates. Tubulin used as a loading control. Densitometry values were normalized to tubulin and WT (n = 2). **e** EMSA of in vitro translated 11-ZF domain of CTCF, full-length CTCF, and BORIS proteins with DNA fragment comprising CTCF/BORIS-bound region at the *Kit* locus. Luciferase protein (Luc) was used as a negative control. The double shift of 11-ZF domain is indicated by black arrows. The experiment was repeated three times. **f** Alignment of human and mouse CTCF/BORIS-bound region for the *Kit* locus. The two 20-bp CTCF motifs were mapped by Motif Alignment and Search Tool. **g** ChIP-seq of CTCF (red) and BORIS (blue) in K562 cells. **h** Western blot of KIT and BORIS in wtK562 and K562 treated with BORIS-specific zinc finger nuclease (ZFN) (n = 2). **i** qPCR analysis of *KIT* expression in wtK562 and K652 treated with ZFN. *KIT* expression is normalized to *GAPDH* (***p value = 4.07e−08, two-sided unpaired *t* test, compared to WT, n = 6). **j** Immunohistochemical detection of KIT in four types of mouse testes. KIT is detected in spermatocytes (SPC) and round spermatids (RS) of CM testis but not in WT (SG spermatogonia). The entire experiment was performed three times. Bars: 100 μm. In the box plots, the lower and upper hinges correspond to the first and third quartiles, the middle line indicates the median. The upper whisker extends from the hinge to the largest value no further than 1.5 times of the interquartile range (IQR). The lower whisker extends from the hinge to the smallest value at most 1.5 times of the IQR.

expression in K562 cells using the zinc finger nuclease approach (Fig. 7i). Upon BORIS downregulation, *KIT* expression was upregulated at least fourfold compared to parental K562 cells (Fig. 7h, i), suggesting the direct involvement of CTCF and BORIS binding in the repression of *Kit* expression in humans as well as in mice.

As mentioned above, KIT expression is normally restricted to primary spermatogonia, where KIT is necessary to trigger the differentiation process in the noncommitted spermatogonia stem cells[61]. We postulated that the persistence of KIT expression beyond spermatogonia could disrupt downstream differentiation of the developing germ cells and lead to the male sterility, as it has been demonstrated in an earlier study[62]. Indeed, immunohisto-chemical analyses of KIT in 3-month-old testes (Fig. 7j) showed an abnormal staining pattern of KIT in CM testis distinct from that in WT, CTCF HET, and BORIS KO mice. In CM testis, in addition to spermatogonia staining, spermatocytes and round spermatids were also highly positive for KIT (Fig. 7j). Therefore,

the increase of KIT expression observed in CM testis can be explained by an atypical upregulation of KIT in meiotic and post-meiotic cells (Fig. 7j), explaining the abnormal sperm phenotypes such as loose and expanded sperm heads and aberrant flagella that are partially or completely coiled around the head (Fig. 1b and Supplementary Fig. 1c). We conclude that loss of BORIS and haploinsufficiency of CTCF in CM testis cause both down-regulation of spermatogenesis-specific genes (Fig. 6) and a "toxic" upregulation of inappropriate genes such as *Kit*, thereby explaining the severe sterility phenotype.

## Discussion

Here we investigated a longstanding question on the mode of functional interaction between CTCF and BORIS in male germ-line. Previously, while BORIS was known to be expressed only in male germ cells, BORIS-null mice inexplicably have only minimal germ cell defects[29,31,32]. Upon generating mice with varied CTCF

and BORIS levels, we uncovered that CTCF can partially compensate for BORIS loss, and when CTCF levels are also reduced, nearly complete loss of fertility occurs. That phenotype evidently stems from the necessity of CTCF and BORIS to heterodimerize at the key 2xCTSes genome wide. Indeed, our analysis shows that a large number of spermatogenesis controlling genes became repressed in CM testis. Such genes typically have heterodimeric CTCF/BORIS-binding sites in their promoters and appear to be directly affected in the CM genotype (Figs. 5d, 6e, and 7a, g and Supplementary Fig. 12a). On the other hand, some genes with heterodimeric CTCF/BORIS sites gain expression. The most intriguing among them is *Kit*, which uncoordinated persistence during germ cell development results in severe morphological and functional abnormalities (Fig. 7). We propose that the complete sterility of CM male mice results from both a failure to activate spermatogenesis-specific genes and the inability to repress pre-meiotic genes in a stage-specific manner. These findings further confirm that CTCF/BORIS heterodimer can be both a transcriptional activator and a repressor at the same time.

Rather unexpectedly, we also showed that CTCF and BORIS cooperation was required for the proper progression through meiosis I. Although the partial depletion of either CTCF in CTCF HET mice or full loss of BORIS in BORIS KO mice had underwhelming impact on male germ cell development, their combined deficiency in CM mice resulted in complete male sterility. The reduced cellularity of seminiferous tubules and the increased apoptosis in meiotic cells of CM testis point to defects in meiotic progression, which could lead to only a small fraction of germ cells progressing to the spermiogenesis stage. Indeed, CM spermatocytes showed an abnormal SC with atypical SYCP3 aggregates at some paired chromosomes and the persistence of unrepaired DSB beyond early pachynema, two processes found to be related in several studies[53,64]. The establishment of the SC itself depends on several factors, including ones that are specific not only for meiotic cells: the formation of DNA loops mediated by somatic and testis-specific cohesin subunits[65], the pairing of autosomal chromosomes[66], and the stage-sensitive repair of DSB[53,67]. Incidentally, each of these processes has been known to be associated with CTCF function in somatic cells[68–72]. However, now we show that CTCF alone is insufficient to mediate similar processes in germ cells and that a cooperation between CTCF and BORIS is required. Our results suggest at least three not mutually exclusive scenarios to explain how a reduction of CTCF expression in the absence of BORIS might lead to defects and/or delays in meiotic progression observed in CM mice. First, CTCF regulates the 3D architecture of eukaryotic genomes by blocking cohesin-mediated extrusion of chromatin loops, anchoring cohesin to CTCF-binding sites[46,73–76]. In CM mice, CTCF occupancy was lost at thousands of genomic regions that could potentially lead to the loss of cohesin occupancy, affecting the formation of meiosis-specific DNA loops, essential for proper recombination during meiosis. Second, CTCF has been reported to be involved in the pairing of X-chromosomes during the X-inactivation in female somatic cells[68,77]. As the pairing of autosomes is a prerequisite for SC formation, the incomplete pairing of autosomes due to the loss of CTCF occupancy may lead to some abnormalities. Third, CTCF facilitates DSB repair via interacting with some DNA repair factors, such as MRE11, CtIP, and Rad51[69,70]. Specifically, the depletion of CTCF profoundly impairs the homologous recombination pathway of DSB repair by blocking CtIP recruitment to repair foci[70]. Thus, the persistence of DSBs beyond early pachynema in CM testis could be related to the deficiency of CTCF. The abnormal formation of the SC was already observed in BORIS KO mice[32], but to a much lesser extent (20% affected cells in BORIS KO versus 40% in CM testis), suggesting a direct involvement of BORIS in SC formation and

the existence of a compensatory mechanism relying on the availability of CTCF.

Most importantly, our results provide compelling evidence that CTCF and BORIS cooperatively control the spermatogenesis-specific transcription program. Indeed, it can be concluded that the infertility of CM mice is primarily due to a disruption of the germ cell-specific transcription, with a pervasive deregulation of a substantial number of genes with known roles in spermatogenesis and sperm development. Undoubtedly, the loss of CTCF and BORIS occupancy at the promoter regions is the main driver of this transcriptional deregulation, especially at the 2xCTS-binding sites normally occupied by both CTCF and BORIS. Such a requirement for the 2xCTS-driven CTCF and BORIS heterodimerization in transcriptional regulation of germline development is vividly illustrated by the two testis-specific genes, *Tsp50* and *Gal3st1*, previously found to be directly regulated by BORIS[29,31,32]. Both genes are essential for spermatogenesis[78,79], and their downregulation upon *Boris* deletion can account for the bulk of subfertility phenotype of BORIS KO mice. The remaining expression of *Tsp50* and *Gal3st1* in BORIS KO mice may be the result of a functional compensation by CTCF for the loss of BORIS. Such a compensatory role for CTCF is consistent with the strikingly severe downregulation of *Tsp50* and *Gal3st1* expression in CM mice. However, we also demonstrated that in CM mice the *Kit* mRNA and protein were aberrantly accumulated in meiotic and post-meiotic cells, where Kit is normally not expressed. While the depletion of either CTCF or BORIS led to a slight but not significant upregulation of Kit expression as compared to WT, the CM mice showed a dramatic upregulation of Kit expression by >12-fold. Similarly, a variety of other spermatogenesis-essential genes (including *Fert2*, *Tssk3*, *Syse2*, *Tcp11*, *Prkar2a*) followed the same pattern: depletion of either CTCF or BORIS only slightly impaired gene expression, but simultaneous ablation of CTCF and BORIS lead to the unexpectedly dramatic deregulation of these target genes. Thus, our data provide convincing evidence of functional cooperation between CTCF and BORIS in male germ cell development. Taking together the male germline-restricted pattern of BORIS expression from sharks to mammals[26,28] and the functional cooperation of CTCF and BORIS at the promoters of germ cell-specific genes, we would suggest that CTCF and BORIS heterodimerization is evolutionarily more adapted in regulation of many germcell-specific transcripts compared to CTCF homodimerization.

In addition to the heterodimerization between CTCF and BORIS on the protein-to-protein level, we also demonstrated that BORIS directly binds the *Ctcf* promoter and stimulates *Ctcf* expression in male germ cells. This was especially apparent in the context of *Ctcf* heterozygosity, where CTCF levels were reduced 75% upon *Boris* deletion compared to WT testis. As a result of such a severe CTCF downregulation in CM testis, CTCF occupancy was lost at thousands of genomic regions, being more dramatic at 2xCTSes than at 1xCTSes. This observation also raises an important mechanistic question: why CTCF occupancy is preferentially lost in the regions bound by both CTCF and BORIS? Based on the data presented in this study and on the published studies, we propose a model to explain the non-additive phenotype of CM mice in comparison to WT, CTCF HET, and BORIS KO mice (Supplementary Fig. 15). This model is based on the well-documented fact that CTCF can exist in the cells in the form of monomers, homodimers, and heterodimers with BORIS[33,45,80]. Furthermore, the transcriptional outcome of CTCF and BORIS heterodimerization is different from CTCF homodimerization at the 2xCTSes[33,43,45], as the two paralogs recruit different protein partners due to their divergence in the N- and C-termini[35]. When CTCF and BORIS are expressed together

in male germ cells, they preferentially form heterodimers at the active promoters and enhancers. This could indicate that CTCF/BORIS heterodimers are more stable and effective at the 2xCTSes as compared to CTCF homodimers at the same sites. The ectopic BORIS expression in BORIS-negative cells provides a telling clue about BORIS KO phenotype in mice. Indeed, upon such expression, 2xCTSes were preferentially co-bound by both CTCF and BORIS instead of CTCF homodimers[33]. Therefore, in a reverse situation in BORIS KO germ cells, it would be logical to expect that CTCF homodimers might physically replace and compensate for the loss of CTCF–BORIS heterodimers due to sufficient supply of CTCF. However, as CTCF protein level in CM testis is so severely reduced, it is likely that such low level of CTCF may be insufficient to form stable homodimers at the 2xCTSes vacated by BORIS. However, the reduced level of CTCF may be sufficient for CTCF monomers to fill the 1xCTSes. In support of this idea, an acute depletion of CTCF using the auxin-induced degron system showed that the CTCF-binding affinity was significantly reduced at the 2xCTSes that require CTCF homodimer binding as compared to 1xCTSes to which CTCF monomers bind[81]. It is also possible that the preferential additional loss of CTCF occupancy at 2xCTSes could be driven by downregulation of some cofactors required for CTCF co-binding or by a biological selection of surviving germ cells. As a result, upon loss of CTCF occupancy at the 2xCTSes, the transcription of target genes is severely affected in CM testis (see proposed model in Supplementary Fig. 15).

We showed that the sterility of CM mice was associated with low sperm count, impaired morphology, and retarded motility of mature spermatozoa. None of these major contributors to male infertility were prominent in CTCF HET or BORIS KO mice. Similar sperm abnormalities and complete sterility were described for *Ctcf*-conditional knockout in pre-leptotene spermatocytes[82]. There the *Ctcf* knockout affected the incorporation of protamine-1 into chromatin and histone retention in mature spermatozoa[82]. During spermiogenesis, protamines replace histones to permit chromatin compaction in the sperm head[83]. However, not all histones are replaced; some genomic regions retain histones to facilitate the transcription of genes essential for early zygotic activation[84–87]. We recently showed that the regions that escape histone-to-protamine replacement are frequently associated with 2xCTSes[33]. Similarly, CTCF has been reported to be associated with the regions of retained histones in sperm[84–86]. As the loss of CTCF occupancy in CM testis preferentially occurred at 2xCTSes, it is likely that the loss of CTCF binding affected the retention of histones at these sites. If so, the loss of CTCF binding would lead to an abnormal sperm owing to the accelerated histone depletion, as was described in the *Ctcf*-conditional knockout[82].

Our analysis of RNA-seq data of four mouse genotypes suggested *Kit* as a candidate gene responsible for CM sperm abnormalities. Indeed, it has been reported before that the expression of a constitutively active *Kit* mutant in spermatids leads to malformations of mature spermatozoa, both in the head and in flagellum[62]. The *Kit* mutant phenotype is strikingly reminiscent of the phenotype of mature spermatozoa in CM mice, suggesting that the aberrantly high expression of KIT in CM spermatids may contribute to the abnormal sperm morphology observed in CM mice.

The present study strongly supports the conclusion of functional cooperation between CTCF and BORIS proteins in the regulation of normal male germ cell development. However, the contributions of CTCF and BORIS to spermatogenesis are not equivalent, as the conditional *Ctcf* knockout results in complete male mice sterility, while *Boris* knockout leads to a subfertility phenotype only. Nonetheless, as we show here there are several processes where both CTCF and BORIS are necessary for a

normal development, including the regulation of structural chromatin processes in meiosis, the control of spermatogenesis-specific gene transcription, and the involvement in sperm genome organization.

The discovery of a well-tuned balance of two paralogous proteins in normal germ cell development contributes to our understanding of the role of aberrant BORIS expression in cancer development and progression. We previously showed that the aberrant BORIS expression in cancer cells is associated with the execution of a germline-like transcription program through CTCF and BORIS heterodimerization at the promoter regions of cancer-testis genes[33,88]. Moreover, ectopic BORIS expression in normal somatic cells has been shown to activate multiple germline-specific transcripts[29,33,89,90]. Thus, by uncovering the vital roles of both CTCF and BORIS for the transcriptional program of normal spermatogenesis, we now can better understand the consequences of aberrant BORIS expression in cancers.

## Methods

**Mice and germ cell preparations**. Boris and Ctcf knockout mouse models have been previously described[22,32]. The compound mutant, $Ctcf^{+/-}Boris^{-/-}$ (CM), mice were generated from mating $Ctcf^{+/-}Boris^{+/+}$ (CTCF HET) females and $Ctcf^{+/+}Boris^{-/-}$ (BORIS KO) males. All mice were genotyped by PCR, as described[22,32]. The use of mice in this study was approved by the NIAID Animal Care and Use Committee under protocol LIG-15. The animals were maintained in the animal facility on a 12-h reverse light/dark cycle and provided with food and water ad libitum. The animal facility was maintained at a temperature of 21 °C with 50–60% humidity.

For germ cell preparations, decapsulated testes of 3-month-old mice were treated with 1 mg/ml of collagenase (Millipore/Sigma) to digest interstitial tissue and release single tubules in suspension. The tubules were next treated with 0.5 µg/ml trypsin (Millipore/Sigma) to generate germ cell-enriched single-cell suspension. The supernatants were then passed through 100 µm cell strainer (Corning) to discard tissue debris and washed with phosphate-buffered saline (PBS). In order to obtain a better cell representation of the phenotypes observed, we included four different mice in each germ cell preparation.

Spermatocytes and round spermatids were purified by centrifugal elutriation as previously described[25] followed by flow cytometric sorting of cells stained with Vybrant DyeCycle Green (Invitrogen, Carlsbad, CA) to obtain cell fractions with high purity. Briefly, decapsulated testes were treated with 1 mg/ml collagenase followed by treatment with 0.5 µg/ml trypsin, and the dissociated cells were used for centrifugal elutriation. Partially purified spermatocytes and round spermatids fractions were incubated with 10 µM Vybrant DyeCycle Green for 30 min at 32 °C followed by 4,6-diamidino-2-phenylindole (DAPI) staining. Cells were then sorted on a FACS-Aria (Becton Dickinson) to purify spermatocytes and round spermatids. DAPI-positive dead cells were eliminated. The purity of cells was confirmed by flow cytometric analysis of DNA content using sorted cells and by cell morphology analysis. The purity of round spermatids fraction was estimated to be approximately 93.7 ± 2.3%. The purity of spermatocytes was estimated to be approximately 81.5 ± 3.7%, based on the immunostaining with anti-SYCP3 antibodies. The expression of genes specific for spermatocytes and round spermatids was confirmed by RNA-seq. The list of marker genes for the specific stages of spermatogenesis was used from Shiota et al.[59].

K562 cell line was grown in Dulbecco's modified Eagle medium supplemented with 10% fetal calf serum and penicillin–streptomycin. K562 *BORIS* knockout cell line was generated previously by us[33].

**Sperm analysis**. The cauda epididymis of 3-month-old mice was dissected and cut into small pieces in PBS and incubated at 37 °C for 15 min allowing the release of spermatozoa to the medium. Sperm number and motility was analyzed in an IVOS system (Hamilton Thorne). To analyze sperm morphology, samples were washed with PBS, fixed in 4% paraformaldehyde, and smeared onto glass slides. Spermatozoa on air-dried slide were stained with hematoxylin.

**Histological analysis, immunohistochemistry, and spermatocyte surface spread staining**. Testes and cauda epididymis were fixed with 10% formalin and embedded in paraffin. Five-micrometer sections were prepared for histological analysis and stained with hematoxylin–eosin. TUNEL staining was performed using the DeadEnd colorimetric TUNEL system (Promega, Madison, WI) according to the manufacturer's protocol. TUNEL-positive cells in 30 tubules were counted to quantify apoptotic cells in testes. Three mice were examined for each genotype.

Surface spreads of spermatocytes were prepared as previously described[44]. Slides were incubated with 0.1% triton X-100 in PBS for 30 min at room temperature and washed 3 times in PBS for 30 min, blocking solution (5% bovine

serum albumin) was added, and slides were incubated for 1 h in a humid chamber at room temperature. Primary antibodies, rabbit anti-γ-H2AX, (ser189) (1:200, Millipore #07-164), mouse anti-SYCP3 (1:50, Santa Cruz sc-74569), and rabbit polyclonal-SYCP1 (1:1000, Abcam #15090), were diluted in blocking buffer, and slides were incubated overnight in a humid chamber at 4 °C. After washing, the following secondary antibodies were used at a 1:200 dilution for 1 h at room temperature: donkey anti-rabbit AlexaFluor488 (Thermo Fisher Scientific A-21206), goat anti-rabbit Texas red (Thermo Fisher Scientific T-2767), donkey anti-mouse AlexaFluor488 (Thermo Fisher Scientific A-21202), and goat anti-mouse Texas red (Thermo Fisher Scientific T-6390). The slides were washed and allowed to dry and mounted with Vectashield Mounting Medium with DAPI (Vector Laboratories). Fluorescent images were acquired using a confocal laser-scanning microscope (LSM 780 Carl Zeiss).

For Kit immunohistochemistry, sections were boiled for 10 min in 10 mM sodium citrate buffer (pH 6.0), treated with 1% $H_2O_2$, blocked with 5% goat serum, and incubated with antibodies against Kit (1:200, anti-Kit AF1356 R&D Systems). The antigen-bound primary antibodies were detected using 1:500 dilution of biotinylated anti-goat IgG antibody (Vector Laboratories BA-5000-1.5) and avidin-conjugated peroxidase (Vector Laboratories). In all immunohistochemical analyses, peroxidase activity was visualized by using 3,3'-diaminobemnzidine as a substrate. Substitution of the primary antibodies with nonimmune IgG was performed as negative control. After staining, sections were counterstained with hematoxylin (MilliporeSigma).

**RNA-seq experiments**. For RNA-seq from whole testis, rRNA-depleted RNA from three independent germ cell preparations described as indicated before (four mice in each preparation) was prepared using the RiboMinus Eukaryote System v2 Kit (Life Technologies) according to the manufacturer's recommendations. In all, 500 ng of rRNA-depleted RNA was used for library preparation using the Ion Total RNA-seq v2 Kit (Life Technologies). The enriched libraries were diluted to a final concentration of 11 pM and subjected to sequencing from a single end in a Ion Proton Sequencer (Life Technologies). FASTQ files were mapped to the UCSC mouse reference genome (build mm9) using two-step alignments. First, the reads were aligned with TopHat2[91]. Second, the unmapped reads from the first step were extracted and aligned with Bowtie2[59] with –local mode and –very-sensitive-local option. For RNA-seq from spermatocytes and round spermatids, three replicates were used for each group, except for WT spermatocytes for which two replicates were used. Total RNA was extracted from cells using Trizol (Life Technologies) according to the protocol provided by the manufacturer. The RNA quality was assessed using an Agilent 2100 Bioanalyzer. The RNA-seq library preparation and sequencing procedures were carried out according to Illumina protocols with minor modifications. Briefly, poly(A)-mRNA was purified from 5 µg of RNA with streptavidin-coated magnetic beads. After chemical fragmentation, mRNA fragments were reverse-transcribed and converted into double-stranded cDNA. Following end repair and A-tailing, paired-end adapters were ligated to the ends of the DNA fragments. The ligated products were amplified with 18 cycles of PCR followed by purification using AMPure beads (Beckman Coulter). The enriched libraries were diluted to a final concentration of 5 nM. Each sample was subjected to 75 cycles of sequencing from a single end in an Illumina HiSeq2000 Sequencer. FASTQ files were mapped to the UCSC mouse reference genome (build mm9) using STAR[92] with default settings. Normalized gene counts and DEGs were obtained with the R package DESeq2[45]. The genes with a $p$ adjusted <0.001 and fold change >2 were considered as significant DEGs. To identify round spermatid marker genes (Supplementary Fig. 9), DEGs between spermatocytes and round spermatids in WT mice were identified using DESeq2 and very stringent conditions to guarantee that we are identifying true markers ($p$ adjusted value <1E−06 and log2Fold change >4). Then the DEGs in CM whole testis were removed from the list for a total of 833 upregulated genes in WT round spermatids compared to WT spermatocytes (post-meiotic markers not affected in CM testis). Additionally, we have used the marker genes for round spermatids from Shiota et al.[59].

GO analysis was performed in DAVID Bioinformatics Resources[93]. GSEA was performed using the GSEA v4.1.0 software[33] (http://software.broadinstitute.org/gsea/index.jsp) and the following Molecular Signatures Database (MSigDB): the Hallmark gene sets (H), the Curated gene sets (C2), and the Ontology gene sets (C5) (http://www.gsea-msigdb.org/gsea/msigdb/collections.jsp). GSEA of the Kit pathway was performed using the Kit Pathway gene set containing 52 genes (https://www.gsea-msigdb.org/gsea/msigdb/cards/PID_KIT_PATHWAY). We chose the $t$ test as metric for ranking genes and 1000 gene set permutations were used to generate a null distribution for the enrichment score, which was used to yield a NES for the gene set.

**ChIP sequencing**. Two germ cell-enriched preparations were obtained as described above. Sixty million cells were crosslinked with 1% formaldehyde for 10 min at room temperature, followed by quenching with 125 mM glycine for 10 min, washed twice with 1× PBS, and resuspended in ChIP lysis buffer (150 mM NaCl, 1% Triton X-100, 0.1% sodium dodecyl sulfate (SDS), 20 mM Tris–HCl pH8.0, 2 mM EDTA). Chromatin was sheared to an average length of 200–500 bp using a Bioruptor sonicator. After overnight incubation with DiaMag magnetic beads (Diagenode, Inc.) and antibodies (4 µg), precipitated chromatin was then washed, decrosslinked, and digested with proteinase K (Millipore/Sigma). The resulting DNA was purified

using phenol/chloroform/isoamyl alcohol. DNA concentration was assessed with the Quant-it PicoGreen dsDNA Kit (Life Technologies) and 5–10 ng was used to generate sequencing libraries. ChIP DNA was amplified using the TruSeq ChIP Sample Preparation Kit (Illumina, Inc., USA). Briefly, the immunoprecipitated material was end-repaired, A-tailed, ligated to the sequencing adapters, amplified by 15 cycles of PCR, and size selected (200–400 bp) followed by single-end sequencing on an Illumina Genome Analyzer according to the manufacturer's recommendations. For spermatocyte ChIP-seq, ~20 million of purified cells were crosslinked with 1% formaldehyde for 10 min at room temperature, followed by the same procedure as described for germ cells extracted from the whole testes. For round spermatids ChIP-seq, we reanalyzed the data previously produced and described by us (GSE70764)[33]. For BORIS ChIP, an antibodies previously characterized by us were used[29,32,94]. For CTCF, an equimolar mix of CTCF C-terminus commercial antibody were used (anti-CTCF B-5 (sc-271514) and C-20 (sc-15914)) from Santa Cruz; anti-CTCF (D31H2) from Cell Signaling; anti-CTCF (07-729) from Millipore; anti-CTCF (70303) from Abcam; anti-CTCF (A300-543A) from Bethyl; and anti-CTCF (NB500-177 and NB500-194) from Novus. For Polymerase II and H3K4me3 ChIP-seq, anti-Pol2-4H8 (Abcam-ab5408) and anti-H3K4me3 (Abcam-ab8580) were used.

Sequences generated by the Illumina Genome Analyzer (50 bp reads) were aligned against the USCS mouse reference genome (build mm9) using Bowtie2 program (http://bowtie-bio.sourceforge.net). The alignment was performed with default parameters using the –m 5 option. Peaks were called using Model-based Analysis for ChIP-Seq (MACS) and (MACS2)[95] using default parameters. For Polymerase II and H3K4me3, the options–nomodel–extsize 74 were included. After MACS, we applied Peak Splitter algorithm (part of MACS) to call sub-peaks, summit of peaks, and improve peak resolution. The ChIP-seq data were visualized using Integrative Genomics Viewer (https://www.broadinstitute.org/software/igv). The peak overlaps between CTCF and BORIS ChIP-seq data sets were carried out with BedTools Suite (http://bedtools.readthedocs.org). We defined peaks as overlapping if at least 1 bp of reciprocal peaks intersect (CTCF/BORIS); the remaining peaks were defined as non-overlapping (CTCF-only and BORIS-only). The normalized tag density profiles were generated using BedTools coverage option from BedTools Suite, normalized to the number of mapped reads, and plotted by Microsoft Excel. The heat maps were generated using seqMINER 1.3.3 platform[96]. We used either $K$-means ranked or linear method for clustering normalization. The summit of either CTCF or BORIS peaks were extended as indicated in the figures. seqMINER was also used to generate the average profiles of read density for different clusters. The annotatePeaks.pl feature of HOMER (http://homer.salk.edu/homer/index.html) was used to generate average plots around CTCF and TSS sites with 10 bp resolution. Default normalization (scaling of tag directories to 10 million reads) was used. Genomic distributions of CTCF and BORIS peaks were identified with the annotatePeaks.pl feature of HOMER(v4.11). CTCF motif analysis were done with MAST (Motif Alignment and Search Tool, MEME suite v5.3.3).

**Western blot**. Protein extracts were prepared by lysing germ cells isolated as describe above ("RNA-seq" section) or K562 cells in RIPA Lysis buffer (Millipore) containing 50 mM Tris–HCl, pH 7.4, 1% Nonidet P-40, 0.25% sodium deoxycholate, 500 mM NaCl, 1 mM EDTA, and 1× protease inhibitor cocktail (Roche Applied Science). Protein samples were resolved by SDS–polyacrylamide gel electrophoresis, transferred to a polyvinylidene difluoride membrane, and incubated with the indicated antibodies. Detections were performed using ECL reagents (Pierce). Kit protein was detected with 1:2000 anti-c-kit (R&D Systems AF1356) and 1:2000 of anti-goat secondary antibody (Santa Cruz sc2020). CTCF protein was detected with 1:2000 anti-CTCF B-5 (Santa Cruz sc-271514) and 1:2000 of an anti-mouse secondary antibody (Santa Cruz sc-2005). Tubulin was detected using 1:5000 anti-α Tubulin B-7 (Santa Cruz sc-5286) and 1:2000 of an anti-mouse secondary antibody (Santa Cruz sc-2005). Images were quantified using the ImageJ (v1.48) software and normalized to Tubulin.

**Electrophoretic mobility shift assay (EMSA)**. DNA fragments encompassing ~200 bp long sequences derived from either CTCF and BORIS ChIP-seq peaks in Kit locus were synthesized by PCR using the following primers 5'-CCACCCAAC TCCGTTTTTTGCAC-3' and 5'-AACGAACACCGTGCGGCTGCAGAG-3'. The PCR products were confirmed by sequencing. EMSA was performed as previously described[94]. Briefly, PCR fragments were labeled using $^{32}P$-γ-ATP with T4 polynucleotide kinase (New England, Biolabs). Protein–DNA complexes were incubated for 1 h at room temperature in binding buffer containing 25 mM Tris pH 7.4, 0.1 mM ZnSO$_4$, 5 mM MgCl$_2$, 5% Nonidet P-40 in PBS, 0.25 mM 2-mercaptoethanol, 10% glycerol, and 0.5 µg of poly dI-dC. Protein–DNA complexes were separated from the unbound probe using 5% native polyacrylamide gels (PAAG) or 1.2% agarose gels run in 0.5× Tris-borate-EDTA buffer. Full-length CTCF, full-length BORIS, and CTCF 11-ZF domain were translated in vitro using the TnT Quick Coupled Transcription/Translation System (Promega).

**Real-time PCR**. Total RNA was prepared using the RNeasy Minikit (Qiagen, Valencia, CA). cDNA was prepared using the SuperScript III first-strand synthesis system (Invitrogen) according to the manufacturer's protocol. Real-time PCR was

performed using the SYBR green PCR master mix (Applied Biosystems, Foster City, CA) and an 7900HT sequence detection system (Applied Biosystems). Expression levels were normalized with the housekeeping gene *Gapdh*. Primers used are listed in Supplementary Data 6.

**Statistical analysis**. In all figure legends, *n* represents the number of independent experiments conducted. Statistical analysis was performed using R. Two-side unpaired *t* test (confidence level 0.95) was used to calculate the *p* value for all figures indicated with the exception of Fig 1e, f where a two-sided unpaired Mann–Whitney's test (confidence level 0.95) was used. Exact *p* values are indicated in the figures or in the figure legends. In box plots, the lower and upper hinges correspond to the first and third quartiles, the middle line indicates the median. The upper whisker extends from the hinge to the largest value no further than $1.5 \times IQR$ from the hinge (where IQR is the interquartile range or distance between the first and third quartiles). The lower whisker extends from the hinge to the smallest value at most $1.5 \times IQR$ of the hinge. In jitter charts (Fig. 1a), the mean is indicated as a horizontal line.

## Data availability
The authors declare that all data that support the findings of this study are available in this article and are provided as a Source file data file or from the corresponding authors upon reasonable request. ChIP-seq and RNA-seq data generated in this study are available on GEO with accession number GSE154249. This study analyzed data from the previously deposited dataset GSE70764. For GSEA [http://www.gsea-msigdb.org/gsea/msigdb/collections.jsp] analysis, the following Molecular Signatures Database (MSigDB) gene sets were used: Hallmark (H), Curated (C2), and Ontology (C5). GSEA of Kit [https://www.gsea-msigdb.org/gsea/msigdb/cards/PID_KIT_PATHWAY] pathway was performed using the Kit Pathway gene set containing 52 genes. Source data are provided with this paper.

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

## Acknowledgements

We thank Dr. Susan Pierce and Dr. Louis Miller for support, critical reading of the manuscript, and helpful discussions. This work was supported by the Intramural Research Program of the National Institute of Allergy and Infectious Diseases (NIAID) and the NIH (both to V.V.L.). The work of J.T.L. was supported by NIH grant R37-GM58839. A.V.S. was supported by the MOST National Key R&D Program of China project number 2018YFA0106903. This study used the Office of Cyber Infrastructure and Computational Biology High Performance Computing cluster at NIAID and high-performance computational capabilities of the Biowulf Linux cluster at NIH.

## Author contributions

Conceived and designed the experiments: S.R.-H., E.M.P., D.L., and V.V.L. Performed the experiments: S.R.-H., E.M.P., S.K., C.F.M.-C., and A.L.K. Analyzed the data: S.R.-H. and E.M.P. Wrote the paper: S.R.-H., E.M.P., A.V.S., J.T.L., and V.V.L. All authors read and approved the final manuscript.

## Funding

## Competing interests

The authors declare no competing interests.
