## [Peer Review File · Nature Communications]

REVIEWER COMMENTS

Reviewer #1 (Remarks to the Author):

Rivera-Hinojosa et al. focus on the functional interplay of mouse CTCF and its paralog BORIS proteins. The latter is a testis-specific factor that probably has evolved to fulfill non-redundant functions with CTCF in male germ cells. The study, therefore, addresses an important question. However, the significance of this study's findings and conclusions are unclear in light of the complex and dynamic cellular composition of studied wild-type and mutant testicular samples. These and other issues are outlined below and should be addressed in a revision:

1. The designation of Ctf^{+/}-; Boris⁻- animals as double knock-outs (DKOs) is misleading. Perhaps it will be better to refer to these as compound mutants (CM).
2. A definitive (but more challenging) experiment would have used a conditional Ctf mutation to eliminate CTCF functions in Boris⁻- spermatogonia using, for example, a widely used STRA8-Cre driver.
3. The authors claim that in some instances (18%), BORIS and CTCF co-occupy the same sites. However, there is no evidence in this study, such as sequential ChIP, to support this claim. The two proteins could bind to the same sites but in different cell types (somatic vs. germ cells or germ cells of different stages). If authors have direct evidence of co-binding of the two proteins in other cell types/experiments, they should refer to those as supporting the possibility of co-binding.
4. The authors do not provide a sufficient description of the germ cell enrichment procedure. Both quantitative and qualitative characteristics of this method are critical for ChIP and RNA-seq studies. It will be essential to determine and report somatic cell content for cell suspensions for all genetic backgrounds. It is hard to ascribe any observed differences in gene expression or site occupancy to studied samples' genetic composition alone without such information. This information is particularly crucial for results concerning the broadly-expressed CTCF protein.
5. Also, there is no record of animals' ages and their respective germ-cell phenotypes. Given the progressive deterioration of the germ cell compartment, all animals should be of similar age and possessing comparable germ cell developmental stages and numbers.

Reviewer #2 (Remarks to the Author):

This is an interesting manuscript that follows up on previous work on the same topic from this same laboratory. The authors describe a tissue-specific member of the CTCF family, BORIS - aka CTCFL, that is expressed uniquely in male germ cells, along with the canonical CTCF. To discern the relative contributions of each member of this gene family, the authors have generated KOs of each gene and expressed these in various combinations of zygosity. They report that while KOs of either gene individually impose only relatively minor effects on spermatogenesis, the double KO of both genes leads to a much more significant impairment of spermatogenesis. This observation plus additional evidence lead the authors to conclude that "a synergistic heterodimeric interaction between CTCF and BORIS is required to both repress pre-meiotic genes and activate post-meiotic genes for a complete spermatogenesis program."

Although this manuscript reports interesting results, there are some concerns. First, there is overlap between the results reported in this manuscript and those previously reported. A 2012 paper by Sleutels et al. (Epigenetics Chromatin 5:8) previously reported the effects of a BORIS/CTCFL KO, and the identification of the binding motif for BORIS/CTCFL was previously reported by the authors of the current manuscript (Rivero-Hinojosa et al., Sci Rep, 7:1-13; 2017). That said, as noted by the authors, the current manuscript reports the only CTCF/BORIS/CTCFL DKO data. However there are additional concerns.

Line 93: The authors suggest that CTCF-BORIS a heterodimer that functions as a developmental activator that is essential for induction of male gametogenesis genes.

Direct evidence in support of this contention is lacking. No data directly showing that the CTCF-BORIS dimer has a function in spermatogenesis is provided. For example, one might have expected that a pull-down of the CTCF complex would have also pulled down BORIS, but no data to this effect is provided. Thus, there is a lack of direct protein-protein interaction data to support the notion that CTCF and BORIS function as a heterodimer.

In general: - The authors suggest that BORIS and CTCF must normally synergize to activate genes critical to the spermatogenic differentiation program. This conclusion is based on comparison of the WT and *Ctcf*^{+/-}*Boris*^{-/-} phenotypes, but would be much better supported with data from comparisons of *Ctcf*^{+/-}*Boris*^{-/-} vs *Ctcf*^{+/+}*Boris*^{-/-} and/or WT vs *Ctcf*^{+/+}*Boris*^{-/-} phenotypes.

Lines 250-253: The total numbers of CTCF peaks is reduced in the DKO. However, there are about 2525 NOVEL peaks that appear in the DKO mouse testis. What is the explanation for the genesis of these new peaks that are not normally found in the WT testis?? How can a *Boris*^{-/-}*Ctcf*^{+/-} DKO produce new peaks?

Lines 623-624: There may be mislabeling in the legend for Figure 3. In the legend, the explanation for part C refers to Venn diagrams, but the Venn diagrams appear in what is currently labeled as part B. Similarly the legend refers to gene tracks in part D, but those currently appear in the figure as part C.

A final potential short-coming of this manuscript is the lack of any data, discussion or even speculation regarding why or how the proposed heterodimer of CTCF and BORIS might function differently than a homodimer of CTCF such that there may have been some sort of evolutionary benefit to conservation of or selection for the expression of a second member of this family in spermatogenic cells.

Reviewer #3 (Remarks to the Author):

Boris, the *Ctcf* gene paralog is expressed in male germ cells in addition to CTCF. Both CTCF and BORIS bind to the same DNA motif and form a heterodimer. Boris knockout male mice have only minor spermatogenesis defects but show sub-fertility.

The important unanswered question is whether CTCF could functionally compensate for the absence of BORIS and whether it could account for the partial phenotypes observed in the Boris ko spermatogenic cells. The present manuscript addresses the functional interplay between CTCF and BORIS following a systematic analysis of male germ cells expressing varying levels of CTCF and BORIS. These are mice with the following genotypes: WT (*Ctcf*^{+/+}*Boris*^{+/+}); CTCF HET (*Ctcf*^{+/-}*Boris*^{+/+}); BORIS KO (*Ctcf*^{+/+}*Boris*^{-/-}); and Double Knock Out (DKO = *Ctcf*^{+/-}*Boris*^{-/-}).

Phenotypical analyses highlighted that DKO male mice are infertile due to severe meiotic defects and large-scale cell apoptosis and the subsequent low sperm count, reduced sperm motility, and abnormal sperm morphology.

Detailed molecular investigations including CTCF and BORIS ChIP-seq mapping, showed that in contrast to the genomic CTCF-binding sites, CTCF/BORIS and BORIS-only was associated with active promoters and enhancers. CTCF ChIP-seq in the *Ctcf*^{+/+}*Boris*^{-/-} mice further suggested a role for BORIS in CTCF-BORIS heterodimer formation at sites of high transcriptional activity. These series of experiments were completed with whole testis transcriptomic analysis to highlight the role of CTCF/BORIS in specific gene expression.

Conclusions and critical points

Overall this manuscript presents an interesting set of data that shed light on the functional interplay between the two paralog factors, CTCF/BORIS. The observations reported are stimulating and could present the basis of interesting additional investigations.

However, there is one important drawback that impacts the solidity of the authors conclusions regarding whole-testis ChIP-seq and transcriptomic analyses. It is of critical importance that the authors completely revise the interpretation of their data regarding the role of CTCF/BORIS in the control of cell transcriptional activity (see below).

Transcriptomic analyses of whole testis from the four tested genotypes shows a significant impact on gene expression mostly in the DKO spermatogenic cells. However, since the testes of DKO mice have a drastically different cell composition, the authors cannot conclude on the role of CTCF/BORIS in gene expression. Indeed, the authors should take into account the fact that different spermatogenic cell populations have different gene expression programs. Therefore, the down-regulated genes could correspond to the absence of cell stage-specific transcripts expressed by the missing cells (i e, protamine mRNA in spermatids) and upregulated genes could be transcripts from the surviving cells that over-contribute in the DKO testes total RNAs.

From the figures presented in the manuscript it is clear that specific sets of spermatogenic cells, particularly post-meiotic cells, are missing in DKO mice testes. Therefore, among the down-regulated genes many should correspond to genes expressed in haploid cells. There are examples in the literature where investigators proposed specific approaches to consider the transcriptional role of a factor independently of the effect of this factor on cell type composition. This approach is exemplified in the case of testis in pmid 30257209 : Fig. S4 (related to Fig. 6). It is based on the principle that a change in the transcriptome due to cell loss should be associated with the downregulation of all the corresponding specific transcripts. If only some of the transcripts are down regulated, while others remained unchanged, then it is possible to conclude that some changes are also due to gene deregulation.

If the above analysis reveals that most of the observed changes is due to a change in cell composition, then the authors could choose an alternative approach.

Indeed, since the authors showed that a significant proportion of the differentially regulated gene are bound by CTCF/BORIS, it is possible to hypothesize that these genes are regulated by CTCF/BORIS.

In any case, the authors should explicitly explain this issue and carefully comment ONLY on transcriptional regulations affecting genes bound by CTCF/BORIS.

All the corresponding data presentations and the discussions concerning the transcriptomic analyses should be revised accordingly (Fig. 6 and S6; text, lanes 346-353; 476-498; 537-539).

The authors are also invited to consider that a loss in a specific cell type, i e, post-meiotic cells, as observed in DKO mice testis, could also explain the loss of CTCF/BORIS on some sites.

They are therefore invited to explicitly and clearly present this possibility.

For the sake of clarity, the text from the reviewers is highlighted in bold, whereas our responses are in italics. References to line numbers in this letter correspond to the manuscript version with changes tracked.

Reviewer #1 (Remarks to the Author):

Rivera-Hinojosa et al. focus on the functional interplay of mouse CTCF and its paralog BORIS proteins. The latter is a testis-specific factor that probably has evolved to fulfill non-redundant functions with CTCF in male germ cells. The study, therefore, addresses an important question. However, the significance of this study's findings and conclusions are unclear in light of the complex and dynamic cellular composition of studied wild-type and mutant testicular samples. These and other issues are outlined below and should be addressed in a revision:

We thank the reviewer for recognizing the importance of our study. To address the question about the complex and dynamic cellular composition of testicular samples, we have added RNA-seq data performed on purified round spermatids and spermatocytes isolated from all four genotypes: Wild-type (WT; Ctcf^{+/+}Boris^{+/+}), Ctcf heterozygous knockout (CTCF HET; Ctcf^{+/-}Boris^{+/+}), Boris knockout (BORIS KO ; Ctcf^{+/+}Boris^{-/-}) and DKO (DKO; Ctcf^{+/-}Boris^{-/-}). We have also added CTCF ChIP-seq data performed on round spermatids and spermatocytes isolated from WT mice to compare CTCF occupancy in whole testes versus individual germ cell types, round spermatids, and spermatocytes. The new data were added in the main Fig. 6, supplementary figures S6, S9, S11, S12 and S14, and described in the main text of manuscript.

1. The designation of Ctcf^{+/-}; Boris^{-/-} animals as double knock-outs (DKOs) is misleading. Perhaps it will be better to refer to these as compound mutants (CM).

We thank the reviewer for pointing out the misleading designation given to Ctcf^{+/-} Boris^{-/-} animals. To follow the recommendation, we have changed the "DKO" to "compound mutant (CM)" throughout the manuscript. We have maintained the DKO designation in this letter for the sake of clarity with other reviewer's comments.

2. A definitive (but more challenging) experiment would have used a conditional Ctcf mutation to eliminate CTCF functions in Boris^{-/-} spermatogonia using, for example, a widely used STRA8-Cre driver.

We agree that this experiment would be interesting, but indeed more challenging. Ctcf conditional knockout in mice by STRA8-Cre recombinase has been generated and described in this study: PMID: 27345455. The Ctcf conditional knockout mice showed some spermatogenic abnormalities similar to what we described for Ctcf^{+/-};Boris^{-/-} compound mutants, including seminiferous tubule atrophy, low sperm counts, abnormal sperm morphology and infertility, but not as severe as in our compound mutants. The comparison of our study with PMID: 27345455 allowed us to conclude that BORIS can partially compensate for the loss of Ctcf. Indeed, in contrast to Ctcf^{+/-}; Boris^{-/-} compound mutants, the Ctcf conditional knockout doesn't show major defects in meiosis I, indicating that the wild-type level of BORIS expression may compensate for the loss of CTCF protein.

However, there are several concerns regarding the mice with conditional Ctcf knockout on the background of Boris knockout. First, as STRA8 gene is a very well-known target of CTCF and BORIS (PMID: 22709888; PMID: 32393311), it might not be the best Cre system for CTCF deletion. Second, as shown before (PMID: 27345455) the conditional deletion of Ctcf using the Stra8-Cre system is not 100% efficient. In Stra8-iCre-Ctcf f/Δ mice a 60% reduction of CTCF protein level in testis extracts was observed (PMID: 27345455). To compare, Ctcf^{+/-}; Boris^{-/-} compound mutants showed 75% reduction of CTCF protein level on the

background of BORIS knockout. Thus, Stra8-iCre-Ctcf f/Δ CTCF conditional knockout combined with Boris-/- knockout may produce exactly the same or similar phenotype as Ctcf+/-; Boris-/- compound mutants.

3. The authors claim that in some instances (18%), BORIS and CTCF co-occupy the same sites. However, there is no evidence in this study, such as sequential ChIP, to support this claim. The two proteins could bind to the same sites but in different cell types (somatic vs. germ cells or germ cells of different stages). If authors have direct evidence of co-binding of the two proteins in other cell types/experiments, they should refer to those as supporting the possibility of co-binding.

CTCF and BORIS heterodimerization in cancer and germ cells has been scrupulously described in our previous study (PMID: 26268681). There, analyzing CTCF and BORIS occupancy in cancer and germ cells we showed that the regions co-bound by both CTCF and BORIS contained at least two closely spaced CTCF binding motifs (termed clustered CTCF Target Sites (CTSes) or 2xCTSes). The 2xCTSes predispose the physical and cooperative interaction between CTCF and BORIS due to DNA-dependent constraints. In the PMID: 26268681 study, we demonstrated the CTCF and BORIS heterodimerization at the 2xCTSes by: 1). sequential ChIP (Chip-Re-ChIP), 2). co-IP of CTCF and BORIS in the presence and absence of DNA, 3). In Situ Proximity Ligation Assay (ISPLA) in cancer and germ cells, 4). Electrophoretic Mobility Shift Assay (EMSA) with K562 nuclear extracts. We also recently demonstrated a stable interaction of CTCF and BORIS in nuclear extracts by EMSA with size-fractionated nuclear extracts (PMID: 31937660). Moreover, recently another group confirmed CTCF and BORIS heterodimerization in neuroblastoma cancer cells resistant to ALK inhibition (PMID: 31391581).

In response to the reviewer's question, we have now added a new supplementary supporting figure, Fig.S4, demonstrating that the sites co-occupied by CTCF and BORIS in mouse testes (CTCF/BORIS sites) contained at least two binding sites under CTCF/BORIS ChIP-seq peaks (new Fig.S4). Moreover, the 2xCTS in the Kit promoter is described in the Fig.7, where we showed the sequence of two closely spaced CTCF binding sites (Fig.7F) and the double occupancy of the target sequence by the 11ZF-DNA binding domain (Fig.7E). We also extended the Introduction section about CTCF and BORIS interactions reported in published studies (lines 85-97) and the Result section describing CTCF and BORIS ChIP-seq data (lines 232-254).

4. The authors do not provide a sufficient description of the germ cell enrichment procedure. Both quantitative and qualitative characteristics of this method are critical for ChIP and RNA-seq studies. It will be essential to determine and report somatic cell content for cell suspensions for all genetic backgrounds. It is hard to ascribe any observed differences in gene expression or site occupancy to studied samples' genetic composition alone without such information. This information is particularly crucial for results concerning the broadly-expressed CTCF protein.

We agree with the reviewer and we have extended the experimental procedure description (lines 16-31, Materials and Methods section). To address the question regarding a contamination with somatic cells or different composition of germ cells between genotypes, we have now purified round spermatids and spermatocytes from all four genotypes and performed RNA-seq and ChIP-seq on these individual germ cell types. These data was compared to the whole testes preparations. Thus, we confirmed the deregulation of spermatogenesis-specific transcription in DKO testes by complementing RNA-seq on whole testes with RNA-seq data on purified round spermatids and spermatocytes. In the new Fig.6B-C,F and the new supplementary figures S9, S11, S12 and S14, we are demonstrating the synergistic impact on the gene expression upon depletion of both CTCF and BORIS in individual germ cells types, round spermatids and spermatocytes, in addition to the whole testis (lines 355-377). Using ChIP-seq data, we compared CTCF occupancy mapped by ChIP-seq in whole testes with CTCF occupancy mapped in purified round spermatids and spermatocytes (Please see the new Fig.S6 and the new lines 296-303). The new Fig.S6 shows that CTCF

occupancy mapped in whole testes is fully maintained in round spermatids and spermatocytes, concluding that CTCF occupancy is generally conserved in the main germ cell types as previously reported for other cell types (PMID 22955616).

5. Also, there is no record of animals' ages and their respective germ-cell phenotypes. Given the progressive deterioration of the germ cell compartment, all animals should be of similar age and possessing comparable germ cell developmental stages and numbers.

All animals used in the study were 3-months old. We have additionally reiterated this info throughout the manuscript.

Reviewer #2 (Remarks to the Author):

This is an interesting manuscript that follows up on previous work on the same topic from this same laboratory. The authors describe a tissue-specific member of the CTCF family, BORIS - aka CTCFL, that is expressed uniquely in male germ cells, along with the canonical CTCF. To discern the relative contributions of each member of this gene family, the authors have generated KOs of each gene and expressed these in various combinations of zygosity. They report that while KOs of either gene individually impose only relatively minor effects on spermatogenesis, the double KO of both genes leads to a much more significant impairment of spermatogenesis. This observation plus additional evidence lead the authors to conclude that "a synergistic heterodimeric interaction between CTCF and BORIS is required to both repress pre-meiotic genes and activate post-meiotic genes for a complete spermatogenesis program." Although this manuscript reports interesting results, there are some concerns. First, there is overlap between the results reported in this manuscript and those previously reported. A 2012 paper by Sleutels et al. (Epigenetics Chromatin 5:8) previously reported the effects of a BORIS/CTCFL KO, and the identification of the binding motif for BORIS/CTCFL was previously reported by the authors of the current manuscript (Rivero-Hinojosa et al., Sci Rep, 7:1-13; 2017). That said, as noted by the authors, the current manuscript reports the only CTCF/BORIS/CTCFL DKO data. However there are additional concerns.

We thank the reviewer for the positive comments and careful review. Indeed, Boris KO mice were described previously in two independent studies (PMID: 22709888 and PMID: 20231363). The binding motif for BORIS was also reported in multiple studies as well (PMID: 26268681, PMID: 28145452). As CTCF and BORIS were shown previously to work as a heterodimer and regulate transcription of multiple cancer-testis genes in cancers (PMID: 26268681, PMID: 31391581, PMID: 31292201, PMID: 16140943), the goal of this study was to compare the phenotype of the single knockouts of either CTCF or BORIS with the DKO of both CTCF and BORIS in mice. This analysis allowed us to make the conclusion regarding the functional cooperation and compensation between two paralogous factors in the regulation of male germ cell development. These data are new and not overlapping with previously published data.

Line 93: The authors suggest that CTCF-BORIS a heterodimer that functions as a developmental activator that is essential for induction of male gametogenesis genes. Direct evidence in support of this contention is lacking. No data directly showing that the CTCF-BORIS dimer has a function in spermatogenesis is provided. For example, one might have expected that a pull-down of the CTCF complex would have also pulled down BORIS, but no data to this effect is provided. Thus, there is a lack of direct protein-protein interaction data to support the notion that CTCF and BORIS function as a heterodimer.

CTCF and BORIS heterodimerization in cancer and germ cells has been previously reported in several studies (for example, PMID: 26268681, PMID: 31391581, PMID: 31292201). CTCF and BORIS form a heterodimer at a subset of binding regions that contain at least two closely spaced CTCF binding motifs (termed clustered CTCF Target Sites (CTSes) or 2xCTSes). As we have already established CTCF and BORIS heterodimerization at the 2xCTSes in PMID: 26268681, in the current manuscript we focused more on the synergistic impact of double depletion of both paralogous factors on the male germ development. Of note, CTCF and BORIS heterodimerization in PMID: 26268681 was confirmed by using 1). sequential ChIP (Chip-Re-ChIP), 2). co-IP of CTCF and BORIS in the presence and absence of DNA, 3). In Situ Proximity Ligation Assay (ISPLA) in cancer and germ cells, and 4). Electrophoretic Mobility Shift Assay (EMSA) with K562 nuclear extracts. In response to both reviewers concerns regarding CTCF-BORIS heterodimerization, we have now added a new supplementary supporting figure demonstrating that the sites co-occupied by CTCF and BORIS in mouse testes (CTCF/BORIS sites) contained at least two binding sites under CTCF/BORIS ChIP-seq peaks (new Fig.S4). Moreover, the 2xCTS in the Kit promoter is described in the Fig.7, where we showed the sequence of two closely spaced CTCF binding sites (Fig.7F) and the double occupancy of the target sequence by the 11ZF-DNA binding domain in the EMSA experiment (Fig.7E). We also extended the Introduction section about CTCF and BORIS interactions reported in published studies (lines 85-97) and the Result section describing CTCF and BORIS ChIP-seq data (lines 232-253).

In general: - The authors suggest that BORIS and CTCF must normally synergize to activate genes critical to the spermatogenic differentiation program. This conclusion is based on comparison of the WT and Ctcf+/-Boris-/- phenotypes, but would be much better supported with data from comparisons of Ctcf+/-Boris-/- vs Ctcf+/+Boris-/- and/or WT vs Ctcf+/+Boris-/- phenotypes.

The goal of this study was to compare all 4 types of genotypes (Ctcf+/+Boris+/+, Ctcf+/-Boris+/+, Ctcf+/+Boris-/- and Ctcf+/-Boris-/-), including the ones suggested by the reviewer. The synergistic phenotype for the male germ cell development was observed and described here only in Ctcf+/-Boris-/- mice compared to the rest of genotypes.

Lines 250-253: The total numbers of CTCF peaks is reduced in the DKO. However, there are about 2525 NOVEL peaks that appear in the DKO mouse testis. What is the explanation for the genesis of these new peaks that are not normally found in the WT testis?? How can a Boris-/-Ctcf+/- DKO produce new peaks?

Indeed, there are 1922 sites where CTCF occupancy increased in DKO compared to WT testes (Fig.5A, Fig.S7), we called them "gained" CTCF sites. Based on the supplementary Fig.S7B-C, the gained CTCF peaks in DKO mice are not entirely "novel" as they are also bound by CTCF in WT testes, but to a lesser extent. The increase of CTCF occupancy in DKO testes could be explained by a change of transcription around these CTCF sites. Of note, a gain of CTCF occupancy in the promoter regions was shown to be generally associated with increased gene expression (for example, in PMID: 32232485). To analyze if this was the case, we overlapped the "gained" CTCF sites with the genomic coordinates of differentially expressed genes in DKO testes, round spermatids and spermatocytes. We found some positive correlation between upregulated genes and gained CTCF peaks in DKO testes, but this was not significant (p-value is 0.75). Another explanation could be either a slower CTCF turnover at these sites to compensate for the lower amount of available CTCF or by the necessity to maintain a high occupancy at these CTCF sites. A sentence regarding those sites has been added in the Results section (lines 308-310).

Lines 623-624: There may be mislabeling in the legend for Figure 3. In the legend, the explanation for part C refers to Venn diagrams, but the Venn diagrams appear in what is currently labeled as part B. Similarly the legend refers to gene tracks in part D, but those currently appear in the figure as part C.

We thank the reviewer for pointing out the mislabeling in Figure 3. We have now corrected this in the figure legend.

A final potential short-coming of this manuscript is the lack of any data, discussion or even speculation regarding why or how the proposed heterodimer of CTCF and BORIS might function differently than a homodimer of CTCF such that there may have been some sort of evolutionary benefit to conservation of or selection for the expression of a second member of this family in spermatogenic cells.

We agree with the reviewer that the outcome of CTCF and BORIS heterodimerization versus CTCF homodimerization at the 2xCTSeqs is an interesting question to discuss. While CTCF and BORIS share a similar 11 ZF-DNA binding domain, they diverge dramatically in their amino and carboxyl termini, resulting in their interaction and recruitment of different protein-partners (PMID: 22168535). In our previous study (PMID: 26268681), using two independent systems modulating BORIS levels in opposite directions, we demonstrated that the functional consequence of either CTCF and BORIS heterodimerization or CTCF homodimerization at 2xCTSeqs is different with respect to the transcriptional regulation of the target genes and loop formation. The fact that CTCF and BORIS heterodimerization is preprogrammed in the sequence of 2xCTSeqs, which are evolutionary conserved and associated mostly with promoters and enhancers (described in detail in PMID: 26268681), already suggests the evolutionary benefits of these interactions between two paralogous factors. There are several studies, showing that ectopic BORIS expression in BORIS-negative cells results in activation of germline-like transcriptional program (PMID: 26268681, PMID: 31292201, PMID: 31391581, PMID: 31292201, PMID: 32393311). The substitution of CTCF homodimer with CTCF and BORIS heterodimer at the 2xCTSeqs is required for the expression of several testis-specific genes as PRSS50, Gal3st1, PRAME, FOXA3, and Stra8. To demonstrate the same in mouse models is a much more challenging task, as multiple stages of spermatogenesis are affected. In the current manuscript we show that multiple testis-specific genes are bound by both CTCF and BORIS and their expression affected by BORIS knockout (BORIS KO mice), but not as significant as in DKO mice (see Fig.6B,C, heatmap in the four types of mice, lines 391-403 and the Discussion section, lines 593-616). This means that CTCF homodimer is able to compensate for the loss of BORIS, but CTCF and BORIS heterodimerization at the promoters of the target genes is more evolutionary adapted in regulation of germ-cell specific transcription. On this matter, we have added a new paragraph into the Discussion section (lines 612-616)

Reviewer #3 (Remarks to the Author):

Boris, the Ctf gene paralog is expressed in male germ cells in addition to CTCF. Both CTCF and BORIS bind to the same DNA motif and form a heterodimer. Boris knockout male mice have only minor spermatogenesis defects but show sub-fertility. The important unanswered question is whether CTCF could functionally compensate for the absence of BORIS and whether it could account for the partial phenotypes observed in the Boris ko spermatogenic cells. The present manuscript addresses the functional interplay between CTCF and BORIS following a systematic analysis of male germ cells expressing varying levels of CTCF and BORIS. These are mice with the following genotypes: WT (Ctcf+/+Boris+/+); CTCF HET (Ctcf+/-Boris+/+); BORIS KO (Ctcf+/+Boris/-); and Double Knock Out (DKO = Ctcf +/-Boris/-). Phenotypical analyses highlighted that DKO male mice are infertile due to severe meiotic defects and large-scale cell apoptosis and the subsequent low sperm count, reduced sperm motility, and abnormal sperm morphology. Detailed molecular investigations including CTCF and BORIS CHIP-seq mapping, showed that in contrast to the genomic CTCF-binding sites, CTCF/BORIS and BORIS-only was associated with active promoters and enhancers. CTCF CHIP-seq in the Ctcf+/+Boris/- mice

further suggested a role for BORIS in CTCF-BORIS heterodimer formation at sites of high transcriptional activity. These series of experiments were completed with whole testis transcriptomic analysis to highlight the role of CTCF/BORIS in specific gene expression.

Conclusions and critical points

Overall this manuscript presents an interesting set of data that shed light on the functional interplay between the two paralog factors, CTCF/BORIS. The observations reported are stimulating and could present the basis of interesting additional investigations. However, there is one important drawback that impacts the solidity of the authors conclusions regarding whole-testis ChIP-seq and transcriptomic analyses. It is of critical importance that the authors completely revise the interpretation of their data regarding the role of CTCF/BORIS in the control of cell transcriptional activity (see below). Transcriptomic analyses of whole testis from the four tested genotypes shows a significant impact on gene expression mostly in the DKO spermatogenic cells. However, since the testes of DKO mice have a drastically different cell composition, the authors cannot conclude on the role of CTCF/BORIS in gene expression.

We agree that the concern raised by the reviewer is valid. Indeed, the DKO adult testis contains multiple populations of germ cells that diverge in their stages of growth and differentiation. In the case of our four types of mice (WT, CTCF HET, BORIS KO and DKO), all of them to some extent produced the final product of spermatogenesis – sperm, meaning that all stages of spermatogenesis and all types of germ cells are present in the testes (Fig.1D and Fig.S1D). Due to apoptosis at all stages of spermatogenesis (Fig.1F), the DKO testes are smaller in size compared to the other mice types (Fig.1C). Thus, we considered that the use of germ cells extracted from whole testis may quickly address the question regarding how spermatogenesis-specific transcriptional programs changed upon depletion of both paralogous factors at all stages of spermatogenesis. The combination of ChIP-seq and RNA-seq allowed us to confirm that the deregulation of transcription is directly linked to CTCF and BORIS binding to the target genes (Fig.6E., Fig.7, Fig.S12). However, we agree with the reviewer that the fluctuation in the proportion of certain germ cell types between the different genotypes may impact the outcome of RNA-seq and ChIP-seq data. Thus, we confirmed the deregulation of spermatogenesis-specific transcription in DKO testes by complementing RNA-seq on whole testes with RNA-seq data on round spermatids and spermatocytes isolated from all 4 types of mice. In the new Fig.6B-C,F and new Fig.S12, we have now not only demonstrated the synergistic impact on the gene transcription upon depletion of both CTCF and BORIS in round spermatids and spermatocytes, but also showed that the genes differentially expressed in round spermatids of DKO mice were deregulated in the same direction in whole testis and vice versa (Fig.6B-C, F and Fig.S11, S12B, S14A), suggesting that both approaches were valid and complemented each other. We have updated the text to reflect those changes (lines 355-377).

Indeed, the authors should take into account the fact that different spermatogenic cell populations have different gene expression programs. Therefore, the down-regulated genes could correspond to the absence of cell stage-specific transcripts expressed by the missing cells (i e, protamine mRNA in spermatids) and upregulated genes could be transcripts from the surviving cells that over-contribute in the DKO testes total RNAs. From the figures presented in the manuscript it is clear that specific sets of spermatogenic cells, particularly post-meiotic cells, are missing in DKO mice testes. Therefore, among the down-regulated genes many should correspond to genes expressed in haploid cells.

The fact that DKO mice produce sperm, although in a much lower quantity compared to WT, CTCF-HET and BORIS KO mice, means that the post-meiotic cells are present in DKO testes, which is also evident from the Fig. 1D and Fig.7J. We agree with the reviewer that not all tubules produce post-meiotic cells, but in these tubules all types of germ cells are missing (Fig.S1D). Following the reviewer's recommendation, we

supplemented the gene expression analysis in testes with the same analysis in single germ cell populations of round spermatids and spermatocytes (new Fig.6C,F, S9, S11, S12, S14), isolated from all 4 types of mice. The new panel C in Fig.6 illustrates that the genes deregulated in post-meiotic round spermatids also showed the deregulation in the same direction in RNA-seq data for whole testes, thus additionally validating our initial conclusions. We have updated the text to reflect those changes (lines 355-377).

There are examples in the literature where investigators proposed specific approaches to consider the transcriptional role of a factor independently of the effect of this factor on cell type composition. This approach is exemplified in the case of testis in pmid 30257209 : Fig. S4 (related to Fig. 6). It is based on the principle that a change in the transcriptome due to cell loss should be associated with the downregulation of all the corresponding specific transcripts. If only some of the transcripts are down regulated, while others remained unchanged, then it is possible to conclude that some changes are also due to gene deregulation.

Following the reviewer's recommendations, we applied the analysis described in PMID 30257209 to our newly generated RNA-seq from round spermatids and spermatocytes. We compared transcriptional profiles in round spermatid and spermatocytes from WT animals and selected a set of markers for post-meiotic cells not affected in DKO testis. Additionally, we also included a subset of markers described in PMID 30257209. The Fig.S9 demonstrates that the expression of markers for SC and RS not only distinguish the two types of germ cells in WT, but also those markers are not significantly affected in DKO germ cells compared to other types of mice, indicating that the transcriptional deregulation observed in DKO testis is a consequence of the depletion of CTCF and BORIS and not due to different cell composition in DKO tubules. Moreover, using RNA-seq from purified round spermatids and spermatocytes, we confirmed the synergistic impact of CTCF and BORIS depletion on the deregulation of multiple genes in DKO mice (new Fig.6B-C, S9, S11, S12, S14). We have updated the text to reflect those changes (lines 355-377).

If the above analysis reveals that most of the observed changes is due to a change in cell composition, then the authors could choose an alternative approach. Indeed, since the authors showed that a significant proportion of the differentially regulated gene are bound by CTCF/BORIS, it is possible to hypothesize that these genes are regulated by CTCF/BORIS.

As indicated above, in the revised manuscript we added the RNA-seq data performed on purified round spermatids and spermatocytes isolated from all four types of mice (Fig.6B-C Fig.S9, S11-12, S14). This allowed us to compare the analysis of RNA-seq data from whole testes with the RNA-seq data from purified SC and RS. Such comparison confirmed that the massive changes in gene transcription are specifically observed only in DKO mice, thus dismissing the concern raised by the reviewer that the changes in transcription are the result of different cell composition.

In any case, the authors should explicitly explain this issue and carefully comment ONLY on transcriptional regulations affecting genes bound by CTCF/BORIS. All the corresponding data presentations and the discussions concerning the transcriptomic analyses should be revised accordingly (Fig. 6 and S6; text, lanes 346-353; 476-498; 537-539).

The main conclusion of our study is that CTCF and BORIS work together in the regulation of spermatogenesis-specific transcriptional programs. Besides the synergistic changes in the germline transcription of DKO mice (Fig.6B,C), we described the multiple examples of direct CTCF and BORIS targets (Fig.6E,F, Fig.7, FigS12) and their deregulation upon loss of BORIS or both CTCF and BORIS . For example, the Fig.7 illustrates the synergistic regulation of the Kit gene by both CTCF and BORIS in DKO mice and in human K562 cancer cell line.

We agree with the comments and we have deleted the part that speculates about transcriptional regulation of differentially expressed genes not bound by CTCF and BORIS (lines 437-447 and 657-661 were deleted in the revised manuscript).

The authors are also invited to consider that a loss in a specific cell type, i e, post-meiotic cells, as observed in DKO mice testis, could also explain the loss of CTCF/BORIS on some sites. They are therefore invited to explicitly and clearly present this possibility.

CTCF occupancy is generally conserved across multiple cell types with only ~20% of binding sites being cell specific (PMID 22955616). To address the reviewer's concern, we compared CTCF occupancy in whole testes with CTCF occupancy in purified round spermatids and spermatocytes using ChIP-seq (new Fig.S6, new lines 293-303). The new Fig.S6 shows that CTCF binding profile in whole testes fully overlaps with CTCF occupancy in RS and SC. Thus, looking on the single examples illustrated in Fig6E, Fig.7A, Fig.S12, one cannot explain the loss of CTCF occupancy at CTCF/BORIS binding sites in DKO testes by the loss of specific cell types, as the major germ cell types, round spermatids and spermatocytes, showed the same CTCF occupancy at these sites.

REVIEWER COMMENTS

Reviewer #1 (Remarks to the Author):

I am satisfied with the authors' responses, additional experiments, and information in the revised manuscript. I therefore would support the publication of this study unless other reviewers pick up on other issues that escaped my attention.

Reviewer #2 (Remarks to the Author):

The authors added some new data and changed other aspects of the manuscript which improved the quality of this work.

The results of this study were that following combined KO of Ctcf and Boris, a large number of spermatogenesis genes lost expression and toxic genes expressed inappropriately. Thus, the authors suggest a synergistic interaction between CTCF and BORIS appears to be required to both repress premeiotic genes and activate postmeiotic genes to properly complete spermatogenesis.

In support of this conclusion, different Ctcf/Boris KO mouse strains were examined by sperm analysis, IHC/ICC and spermatocyte surface spread staining; RNA-seq, ChIP-seq of spermatocytes and spermatids; western blot and qRT-PCR. In addition, previously published data were integrated into this analysis. However, some significant concerns remain including the following:

1. Expression of CTCF protein was not fully eliminated in mouse spermatogenic cells. Thus, as pointed out by Reviewer 1, a more complete ablation of CTCF expression would facilitate more definitive conclusions regarding its role during spermatogenesis. Specifically, in the CTCF HET strain described in Fig. 4C,D, it is not possible to definitively determine the extent to which the influence of CTCF may still impact the function of BORIS. In turn, it is difficult to definitively determine the extent of the functional role contributed by BORIS itself on the basis of these experiments. As shown in past studies (PMID: 27345455) the conditional deletion of Ctcf using the Stra8-Cre system is not 100% efficient. In this respect, the authors' conclusion that "the wild-type level of BORIS expression may compensate for the loss of CTCF protein" remains somewhat speculative.
2. The authors continue to suggest that CTCF and BORIS function synergistically, but evidence in support of this contention remains largely lacking. There are distinct binding sites ascribed to BORIS and CTCF, suggesting they carry out different functions. No direct evidence is provided to definitively demonstrate that these factors function synergistically.
3. Figure 3B describes 1397 BORIS-unique peaks and 6074 CTCF-BORIS- peaks. How do these compare with the CTCF peaks described for the CM mouse testis in Figure 5A, or with the BORIS binding patterns in the Ctcf HET mouse testis? Are the authors suggesting that BORIS compensates for the reduced CTCF levels in the Ctcf HET mouse testis? Finally, how do the binding patterns above compare to those for CTCF in the Ctcf KO mouse testis? If CTCF and BORIS function synergistically, theoretically, the Boris KO should impact CTCF binding patterns, but it is not clear from the data presented that this is the case.
4. Beyond the lack of definitive evidence indicating that CTCF and BORIS function synergistically during spermatogenesis, there is also a lack of even a suggested hypothetical molecular mechanism by which this might happen. That spermatogenesis is clearly compromised in Ctcf cKO or Boris cKO mouse testes has been well documented previously. The present study aims to extend these observations to those associated with the effects of a combined Ctcf+/-Boris-/- mutant, but definitive conclusions about the mechanistic impact of the observations associated with this new mutant phenotype are difficult to support based on the evidence provided, particularly with respect to the specific mechanism by which these two factors might act synergistically to regulate spermatogenesis. This study has profiled the phenotypes in Ctcf+/-Boris-/- mouse testis, but has not revealed definitive mechanistic insight. In this respect, this manuscript still seems lacking.

Reviewer #3 (Remarks to the Author):

Most of the concerns raised previously have been taken into account and the manuscript has now been much improved. I can therefore recommend this manuscript for publication.

Saadi Khochbin

Reviewer #1 (Remarks to the Author):

I am satisfied with the authors' responses, additional experiments, and information in the revised manuscript. I therefore would support the publication of this study unless other reviewers pick up on other issues that escaped my attention.

We thank the Reviewer#1 for supporting the publication of our study. We greatly appreciate all valuable comments and suggestions, which helped us to improve the quality of the manuscript.

Reviewer #2 (Remarks to the Author):

The authors added some new data and changed other aspects of the manuscript which improved the quality of this work.

The results of this study were that following combined KO of Ctf and Boris, a large number of spermatogenesis genes lost expression and toxic genes expressed inappropriately. Thus, the authors suggest a synergistic interaction between CTCF and BORIS appears to be required to both repress premeiotic genes and activate postmeiotic genes to properly complete spermatogenesis.

In support of this conclusion, different Ctf/Boris KO mouse strains were examined by sperm analysis, IHC/ICC and spermatocyte surface spread staining; RNA-seq, ChIP-seq of spermatocytes and spermatids; western blot and qRT-PCR. In addition, previously published data were integrated into this analysis. However, some significant concerns remain including the following:

1. Expression of CTCF protein was not fully eliminated in mouse spermatogenic cells. Thus, as pointed out by Reviewer 1, a more complete ablation of CTCF expression would facilitate more definitive conclusions regarding its role during spermatogenesis. Specifically, in the CTCF HET strain described in Fig. 4C,D, it is not possible to definitively determine the extent to which the influence of CTCF may still impact the function of BORIS.

It is important that we be crystal clear that the reviewer's suggestion to ablate CTCF is not possible. CTCF is an essential protein which supports the 3D genome organization in all vertebrates. The complete ablation of CTCF expression results in a cell death, as it has been documented in multiple studies (PMID: 22532833; PMID: 30513694). Thus, it would be impossible to address any specialized role of CTCF during spermatogenesis on the background of massive apoptosis and cell death in cells with a complete knockout of CTCF.

In turn, it is difficult to definitively determine the extent of the functional role contributed by BORIS itself on the basis of these experiments. As shown in past studies (PMID: 27345455) the conditional deletion of Ctf using the Stra8-Cre system is not 100% efficient. In this respect, the authors' conclusion that "the wild-type level of BORIS expression may compensate for the loss of CTCF protein" remains somewhat speculative.

Indeed, we do discuss the possibility that BORIS may compensate for the loss of CTCF in Ctcf conditional knockouts by comparing our results in Ctcf^{f/+}-Boris^{-/-} mice with the mentioned published study (PMID: 27345455). We discuss our results in relation to CTCF conditional knockout mice, which doesn't show any major defects in meiosis I in contrast to multiple meiotic defects described in Ctcf^{f/+}-Boris^{-/-} mice. This sentence is in the Discussion section, where it is appropriate for us to suggest such comparison and take the liberty to speculate.

2. The authors continue to suggest that CTCF and BORIS function synergistically, but evidence in support of this contention remains largely lacking. There are distinct binding sites ascribed to BORIS and CTCF, suggesting they carry out different functions. No direct evidence is provided to definitively demonstrate that these factors function synergistically.

The major finding of our study is the sterility of compound mutant mice (Ctcf^{f/+}-Boris^{-/-}) in which the reduction in CTCF and deficiency in BORIS proteins resulted in the sterility phenotype, not observed with the single knockouts (Ctcf^{f/+}-Boris^{+/+} or Ctcf^{+/+}Boris^{-/-}). The novel phenotype of Ctcf^{f/+}-Boris^{-/-} mice supports the cooperativity of CTCF and BORIS in germ cells, as do other additional data including the observations of CTCF and BORIS co-binding to the promoters of testis-specific genes (Fig. 3,S4), CTCF and BORIS co-regulation (Fig.4) and their cooperative regulation of transcriptional program (Fig.6,7,S8,S10,S11, S12, S14).

However, it is clear that the word 'synergy' could be confusing and as commonly used, may not be quite right to describe the functional relationship between CTCF and BORIS. Consequently, to avoid such confusion we made the changes throughout the whole manuscript to reflect that CTCF and BORIS function cooperatively producing a synergistic effect in the phenotype of CM mice.

Regarding the second question, CTCF and BORIS have been demonstrated to have both overlapping and not overlapping functions, similar to their binding patterns. For example, BORIS, in contrast to CTCF, is not able to stop cohesin extrusion (PMID: 31937660) and is not involved in 3D genome organization (PMID: 32393311). As it is impossible in the scope of one study to address all aspects of CTCF and BORIS functions during spermatogenesis, we focused more on the function of overlapping CTCF and BORIS binding sites. These CTCF/BORIS sites are enriched at the active promoters and enhancers, mapped in the male germ cells (Fig.3D-E). These sites were affected more upon loss of CTCF occupancy in CM mice (Fig.5). Moreover, the loss of these overlapping sites resulted in the changes of gene expression (Fig.6E, Fig.7, S12A) and related pathways (Fig.S10). The most striking example is the KIT gene (Fig.7, S14), which is a direct evidence of CTCF and BORIS functioning cooperatively to regulate Kit expression during all steps of spermatogenesis.

3. Figure 3B describes 1397 BORIS-unique peaks and 6074 CTCF-BORIS- peaks. How do these compare with the CTCF peaks described for the CM mouse testis in Figure 5A, or with the BORIS binding patterns in the Ctcf HET mouse testis? Are the authors suggesting that BORIS compensates for the reduced CTCF levels in the Ctcf HET mouse testis? Finally, how do the binding patterns above compare to those for CTCF in the Ctcfl KO mouse testis? If CTCF and BORIS function synergistically, theoretically, the Boris KO should impact CTCF binding patterns, but it is not clear from the data presented that this is the case.

To address the reviewer comments, we changed the panel A of Fig.S7 to show an overlap of data from Fig.3B and Fig.5A. This figure additionally demonstrates that CTCF can compensate for the loss of BORIS in BORIS knockout cells: 28 BORIS-only binding sites in WT testis are occupied by CTCF upon loss of BORIS in CM testis. We did not analyze the BORIS binding pattern in CTCF HET mice as we expected the pattern to be similar to that from WT mice: CTCF HET mice have phenotype similar to WT in respect to sperm production (Fig.1) and gene expression (Fig.6). The level of CTCF protein in CTCF HET is similar to WT mice (Fig.4C,D), thus we do not expect BORIS to compensate for the reduced level of CTCF in CTCF HET. Regarding CTCF binding pattern in BORIS KO testis, we compared CTCF occupancy in round spermatids and spermatocytes of BORIS KO and WT mice. Overall, we found that CTCF occupancy is not affected by BORIS depletion in BORIS KO mice compared to WT mice. These data were not included in the manuscript. BORIS may or may not impact CTCF binding pattern, which is a subject of ongoing research.

4. Beyond the lack of definitive evidence indicating that CTCF and BORIS function synergistically during spermatogenesis, there is also a lack of even a suggested hypothetical molecular mechanism by which this might happen. That spermatogenesis is clearly compromised in Ctcf cKO or Boris cKO mouse testes has been well documented previously. The present study aims to extend these observations to those associated with the effects of a combined Ctcf^{+/-}-Boris^{-/-} mutant, but definitive conclusions about the mechanistic impact of the observations associated with this new mutant phenotype are difficult to support based on the evidence provided, particularly with respect to the specific mechanism by which these two factors might act synergistically to regulate spermatogenesis. This study has profiled the phenotypes in Ctcf^{+/-}-Boris^{-/-} mouse testis, but has not revealed definitive mechanistic insight. In this respect, this manuscript still seems lacking.

We respectfully disagree with the reviewer that our study described only the phenotype of Ctcf^{+/-}-Boris^{-/-} mouse testis, but not the mechanism. The mechanism is the cooperation of CTCF and BORIS in the regulation of spermatogenesis expression program. First of all, we show that the infertility of CM mice is primarily due to a disruption of the germ cell specific transcription, with a pervasive deregulation of a substantial number of genes with known roles in spermatogenesis and sperm development (Fig.6,7; S8, S10, S11-14). Second, we show that the loss of CTCF and BORIS occupancy at the promoter regions is the main driver of this transcriptional deregulation (Fig.6, Fig.7, S12). Third, we show multiple examples of such target genes, and their dependence on CTCF and BORIS co-binding to their promoters. For example, we demonstrated that the Kit gene, upon loss of both CTCF and BORIS binding in Ctcf^{+/-}-Boris^{-/-} testes, became aberrantly activated in the late stages of spermatogenesis, where it is not supposed to be expressed during normal spermatogenesis (Fig.7). In fact, the aberrant presence of the KIT protein in meiotic and post meiotic cells is already sufficient to explain the sterility phenotype of Ctcf^{+/-}-Boris^{-/-} mice, as it has been documented before in PMID: 15736269. Moreover, it has been shown in numerous published studies that CTCF and BORIS execute a germline-like transcription program in cancers, where BORIS is aberrantly activated and co-binds with CTCF to the promoters of cancer-testis genes. Thus, all of the above is the mechanism explaining the sterility phenotype of Ctcf^{+/-}-Boris^{-/-} mice.

Reviewer #3 (Remarks to the Author):

Most of the concerns raised previously have been taken into account and the manuscript has now been much improved. I can therefore recommend this manuscript for publication.

Saadi Khochbin

We thank Dr. Saadi Khochbin for recommendation of our study for publication. We greatly appreciate all his valuable comments and critiques, which have been very helpful in improving the quality of our manuscript.

REVIEWER COMMENTS

Reviewer #2 (Remarks to the Author):

The authors have made some effort to respond to my most recent concerns. In some cases I agree that certain points cannot be definitively resolved. However in that context it is important that this is acknowledged in the text. In some cases the authors have endeavored to revise the text – e.g. changing the wording from CTCF and BORIS functioning “synergistically” to functioning “cooperatively” though qualifying the latter with the subsequent statement “producing a synergistic effect” minimizes the extent of this revision. In other cases the authors provide rebuttal comments in their response to the reviewers’ comments but it is not clear the same points are noted or acknowledged in the text.